# Efficient catalyst-free N$_2$ fixation by water radical cations under ambient conditions

Xiaoping Zhang[1], Rui Su[2,3], Jingling Li[1], Liping Huang[2], Wenwen Yang[1], Konstantin Chingin[2], Roman Balabin [2], Jingjing Wang[2], Xinglei Zhang[1], Weifeng Zhu[2], Keke Huang[3], Shouhua Feng[3] & Huanwen Chen [1,2] ✉

The growth and sustainable development of humanity is heavily dependent upon molecular nitrogen (N$_2$) fixation. Herein we discover ambient catalyst-free disproportionation of N$_2$ by water plasma which occurs via the distinctive HONH·HNOH$^{+\bullet}$ intermediate to yield economically valuable nitroxyl (HNO) and hydroxylamine (NH$_2$OH) products. Calculations suggest that the reaction is prompted by the coordination of electronically excited N$_2$ with water dimer radical cation, (H$_2$O)$_2^{+\bullet}$, in its two-center-three-electron configuration. The reaction products are collected in a 76-needle array discharge reactor with product yields of 1.14 µg cm$^{-2}$ h$^{-1}$ for NH$_2$OH and 0.37 µg cm$^{-1}$ h$^{-1}$ for HNO. Potential applications of these compounds are demonstrated to make ammonia (for NH$_2$OH), as well as to chemically react and convert cysteine, and serve as a neuroprotective agent (for HNO). The conversion of N$_2$ into HNO and NH$_2$OH by water plasma could offer great profitability and reduction of polluting emissions, thus giving an entirely look and perspectives to the problem of green N$_2$ fixation.

Nitrogen is an essential element for all living organisms on our planet. Molecular nitrogen (N$_2$) accounts for more than 99% of global nitrogen[1] but is chemically stable (N≡N bond energy ca. 9.5 eV) and thus cannot be directly utilized unless fixed by alternating its oxidation state into bioavailable forms[2]. The problem of N$_2$ fixation is one of the most important for sustainable chemistry. Currently, N$_2$ on Earth is predominantly fixed through geochemical processes such as lightning, biologically by nitrogenases, and industrially through the Haber-Bosch (HB) process[3]. While HB is currently the major industrial process for N$_2$ fixation to ammonia (NH$_3$), it is associated with intensive reaction conditions (ca. 100 bar and 500 °C), severe environmental pollution (>1% of the global carbon emission) and high consumption of fossil fuel (1%–2% of global energy consumption). These issues are becoming increasingly crucial for the global sustainable development, and urge alternative strategies for N$_2$ fixation under mild conditions[4,5]. Extensive research is being done in search of alternative strategies for N$_2$ fixation, including electrocatalytic[6,7], photocatalytic[8], biological[9], and plasma-based[10] methods, but none of these methods have yet been able to rival the overall performance of HB process with regard to the cost efficiency, scalability and selectivity of N$_2$ fixation[6,11,12].

Recent studies indicate that the N≡N bond can be weakened by accepting electrons from the bonding orbitals of N$_2$ to the antibonding orbitals and/or donating electrons, which would make its functionalization feasible[13,14]. The weakening of N≡N bond could be further promoted through the excitation of N$_2$ into its triplet state, e.g., by electronic or collisional activation with molecules or ions[15,16]. Recently, we have discovered that abundant radical cations of water clusters, especially in the dimer form, (H$_2$O)$_2^{+\bullet}$, can be produced by electron stripping from neutral water molecules in a strong electric field of the energy-tunable corona discharge of the pure water vapor at atmospheric pressure[17]. The as-prepared water radical cations showed the high reactivity toward a wide range of volatile molecules, such as benzene, carbon-carbon double bond, acetone, ethyl acetate and dimethyl disulfide, revealing rich chemistry with the ionic and radical characters[17–21].

[1]Jiangxi Key Laboratory for Mass Spectrometry and Instrumentation, East China University of Technology, Nanchang 330013, P. R. China. [2]School of Pharmacy, Jiangxi University of Chinese Medicine, Nanchang 330004, P. R. China. [3]State Key Laboratory of Inorganic Synthesis and Preparative Chemistry, College of Chemistry, Jilin University, Changchun 130012, P. R. China. ✉e-mail: chw8868@gmail.com

In this work, we discovered that, owing to its distinct two-center-three-electron (2c-3e) configuration, $(H_2O)_2^{+\bullet}$ can specifically activate $N_2$ via the formation of HONH·HNOH$^{+\bullet}$ intermediate to selectively disproportionate it into hydroxylamine ($NH_2OH$) and nitroxyl (HNO) products under mild ambient conditions and with no catalyst involved. These products are not commonly observed upon $N_2$ fixation and have high value. $NH_2OH$ is widely used in medicine, textile industry, electronics, chemical synthesis, nuclear industry and other fields[22]. HNO is a valuable material for medical and biology utilities, particularly for biological targets of thiols and metalloproteins in fighting cancer[23]. Importantly, HNO is known for its cardioprotective and neuroprotective effects and is resistant to superoxide scavenging and tolerance development[24]. Overall, the study provides an interesting twist on $N_2$ fixation in terms of both the approach and the products.

## Results and discussion

### Disproportionation reaction of $N_2$ with $(H_2O)_2^{+\bullet}$

In our first experiment, water plasma was generated by discharge of water/argon vapor mixture (Supplementary Fig. 1a). The major ionic species observed by real-time mass spectrometry (MS) detection included protonated water clusters, $(H_2O)_2H^+$ ($m/z$ 37) and $(H_2O)_3H^+$ ($m/z$ 55), as well as abundant water dimer radical cation, $(H_2O)_2^{+\bullet}$ ($m/z$ 36) (Supplementary Fig. 1b), in agreement with previous reports[17–20,25,26]. Remarkably, when neutral $N_2$ was introduced to intersect with the water plasma ca. 1 cm away from the discharge area (Supplementary Fig. 1c), abundant ions at $m/z$ 32 and $m/z$ 33 and $m/z$ 64 were observed (Supplementary Fig. 1d). This observation indicates that the signals $m/z$ 32, $m/z$ 33 and $m/z$ 64 correspond to the species formed due to the interaction between water plasma and neutral $N_2$. When $N_2$ was directly flown into the discharge area together with water vapor through the same channel (Fig. 1a), the same product ions at $m/z$ 32 and $m/z$ 33 and $m/z$ 64 were observed with ca. two-fold higher intensity (Fig. 1b). The higher intensity of product ions in the single-channel configuration is probably due to the higher density of $N_2$ and water plasma right at the end of the electrode than ca. 1 cm away from the end of the electrode (as in the two-channel configuration), which results in higher collision rate between $N_2$ and water plasma species. Therefore, the single-channel configuration was applied in the subsequent scale up experiments to obtain higher yields of products. We tentatively assigned these signals to HNOH$^+$ ($m/z$ 32), $NH_2OH^{+\bullet}$ ($m/z$ 33) and HONH·HNOH$^{+\bullet}$ ($m/z$ 64), respectively. We speculated that these ionic species could be derived through the reaction between $(H_2O)_2^{+\bullet}$ and $N_2$. To verify this assumption, $(H_2O)_2^{+\bullet}$ ions ($m/z$ 36) formed in the water plasma were selectively isolated in the ion trap by radio frequency (RF) field with peak-to-peak voltage of ~ 100 V in the presence of neutral $N_2$ gas, which was directly introduced into the ion trap (Fig. 1c). This experiments design allowed us to specifically probe the intrinsic reactivity of $N_2$ toward $(H_2O)_2^{+\bullet}$ in vacuum without any chemical interference. When the trapped $(H_2O)_2^{+\bullet}$ ions were activated by RF field, the product signals were unambiguously observed at $m/z$ 18 (elimination of $H_2O$ from $(H_2O)_2^+$), $m/z$ 19 (elimination of ·OH from $(H_2O)_2^+$), $m/z$ 33 ($NH_2OH^{+\bullet}$), $m/z$ 51 ($NH_2OH^{+\bullet}$ plus $H_2O$), $m/z$ 55 ($(H_2O)_3H^+$) and $m/z$ 64 (HONH·HNOH$^{+\bullet}$) (Fig. 1d). These observations confirm the occurrence of reaction between $(H_2O)_2^{+\bullet}$ and $N_2$. Curiously, the signal of HNOH$^+$ abundantly produced upon the interaction between water plasma and $N_2$ ($m/z$ 32 in Fig. 1b) was almost undetectable when $(H_2O)_2^{+\bullet}$ was activated in the ion trap (Fig. 1d), probably because the HNO was initially created as neutral species. During the interaction between water plasma and $N_2$, neutral HNO could be easily protonated by other ionic species such as $(H_2O)_2H^+$ ($m/z$ 37) to give the protonated signal at $m/z$ 32 (Fig. 1b), whereas when produced in the ion trap neutral HNO would be instantly pumped out of the ion trap (maintained at high vacuum of $10^{-5}$ Torr).

The signal at $m/z$ 64 was assigned to HONH·HNOH$^{+\bullet}$ intermediate produced during the reaction between $N_2$ and $(H_2O)_2^{+\bullet}$.

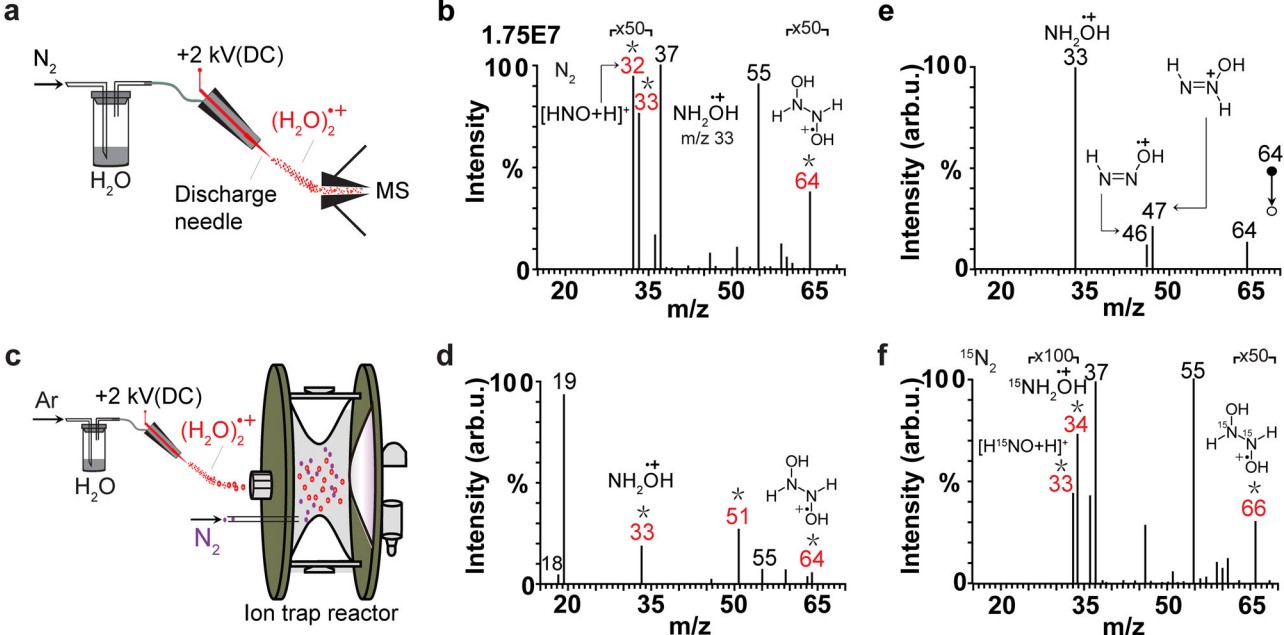

**Fig. 1 | Disproportionation reaction of $N_2$ with water dimer radical cation.**
**a** Experimental setup to study the interaction of $N_2$ with water vapor plasma at ambient conditions. Stainless-steel discharge needle was used as electrode. DC: direct current. The figure is adapted with permission from refs. 18,20,21. **b** The corresponding mass spectrum of ionic products in Fig. 1a. Asterisks correspond to the products specific to the reaction between water vapor plasma and $N_2$. **c** Ion trap reactor applied to study the reaction between $N_2$ and isolated $(H_2O)_2^{+\bullet}$ ($m/z$ 36) in vacuum. The figure is adapted with permission from refs. 18,20,21. **d** The corresponding mass spectrum of ionic products in **c**. Asterisks correspond to the products specific to the reaction between water vapor plasma and $N_2$. **e** Ionic fragments of the reaction intermediate at $m/z$ 64 induced by collisional activation inside the ion trap. **f** Mass spectrum of the ionic species observed during the interaction between water vapor plasma and $^{15}N_2$ ($^{15}N_2$ gas instead of $^{14}N_2$ in **a**). Asterisks correspond to the products specific to the reaction between water vapor plasma and $^{15}N_2$.

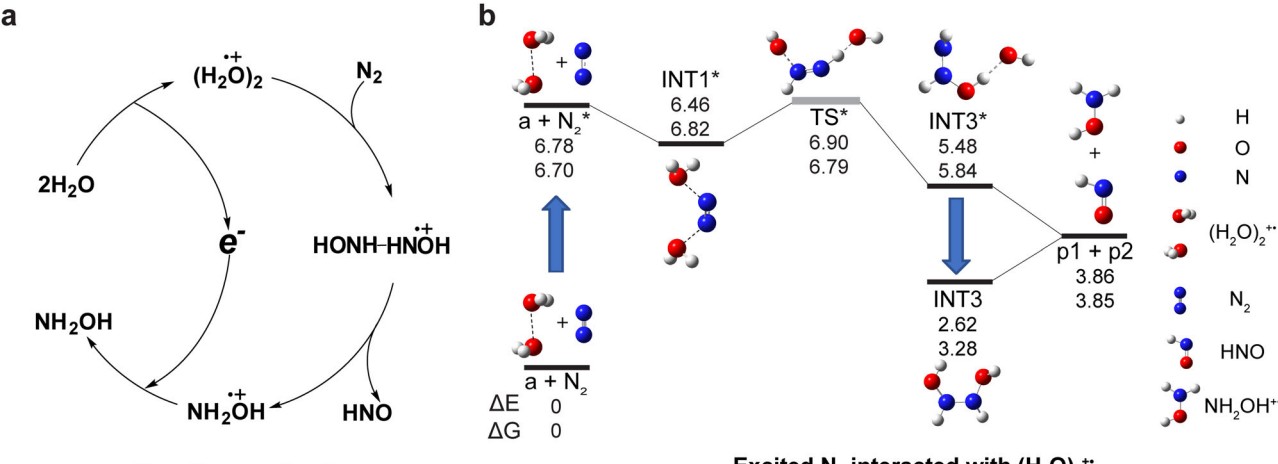

**Fig. 2 | Mechanism and calculation results for the reaction of $N_2$ with $(H_2O)_2^{+\bullet}$.** **a** Schematic diagram summarizing a possible mechanism for the reaction of $N_2$ with $(H_2O)_2^{+\bullet}$. **b** The geometries and energies (in eV at 298 K and 1 atm pressure) of possible molecular and ionic species involved in the disproportionation reaction $N_2 + (H_2O)_2^{+\bullet} \rightarrow NH_2OH^{+\bullet} + HNO$ calculated with CCSD(T) method. Our expected accuracy is 0.04 eV, with the exception of transition state (TS) structure (gray, see Supplementary Note 1). Vertical arrows correspond to the process of electronic excitation/de-excitation. a: $[H_2O\bullet\bullet\bullet OH_2]^+$. p1: $NH_2OH^{+\bullet}$. p2: HNO. The atomic coordinates of the optimized computational models are shown in Supplementary Data File.

The chemical assignment of the *m/z* 64 signal to HONH·HNOH$^{+\bullet}$ was supported by collision-induced dissociation (CID) data (Fig. 1e), showing the characteristic fragment at *m/z* 33 corresponding to the elimination of HNO, accompanied by lower-intensity fragments at *m/z* 46 and *m/z* 47 (also discerned in the spectra Fig. 1b), corresponding to the elimination of $H_2O$ and $^\bullet$OH (the dissociation path shown in Supplementary Fig. 1e), respectively. These chemical assignments were further supported by experiments with isotopic substituents (Fig. 1f, Supplementary Fig. 1f, g). When $H_2O$ in our experiments was replaced by deuterated water, $D_2O$, the abundant signals at *m/z* 33, *m/z* 34 and *m/z* 35 (Supplementary Fig. 1f) were observed, indicating the formation of HNOD$^+$, DNOD$^+$/NH$_2$OD$^{+\bullet}$, and NHDOD$^{+\bullet}$, respectively. Similarly, when $N_2$ was replaced by $^{15}N_2$, the signals at *m/z* 33 and *m/z* 34 were detected (Fig. 1f), indicating the formation of H$^{15}$NOH$^+$ and $^{15}$NH$_2$OH$^{+\bullet}$, respectively. Interestingly, the HONH·HNOH$^{+\bullet}$-type intermediate was also detected in the experiments with isotopic substitutes: as DOND·DNOD$^{+\bullet}$ (*m/z* 68) for the $D_2O$ experiment (Supplementary Fig. 1f) and as HO$^{15}$NH·H$^{15}$NOH$^{+\bullet}$ (*m/z* 66) for the $^{15}N_2$ experiment (Fig. 1f), respectively. Note that the intermediates labeled with different isotopes showed the identical fragmentation path (Supplementary Fig. 1g). Further reference experiments carried out in our lab indicated that the observed nitrogen disproportionation occurred specifically as the result of reaction between neutral $N_2$ and $(H_2O)_2^{+\bullet}$: no products could be detected when isolated $N_2^{+\bullet}$ was exposed to neutral water vapor in the ion trap (Supplementary Fig. 1h). Therefore, the spectral data obtained in all the above-mentioned experiments strongly indicate that the observed species correspond to the disproportionation reaction of $N_2$ with $(H_2O)_2^{+\bullet}$.

**Reaction mechanism**

Derived from the experimental observations and theoretical calculations (as detailed in Supplementary Data File), a possible reaction pathway for $N_2$ disproportionation with $(H_2O)_2^{+\bullet}$ is proposed as shown in Fig. 2a. At the first step, neutral $H_2O$ is ionized to form $(H_2O)_2^{+\bullet}$ species[17–20]. Our calculations indicate that $N_2$ disproportionation with $(H_2O)_2^{+\bullet}$ is thermodynamically not allowed ($\Delta E \approx 3.8$ eV) when $N_2$ is present in its ground singlet (X $^1\sum_g^+$) state (Supplementary Fig. 2) but may occur ($\Delta E \approx -2.9$ eV) when $N_2$ is present in its more active triplet (A $^3\sum_u^+$) state, $N_2^*$ (Fig. 2b). It is well known that $N_2$ is effectively transferred from its singlet state to its

triplet state through the collisions with electrons (*e.g.*, in $N_2$ and $CO_2$ gas lasers), ions or molecules[15]. It has been shown that, owing to its high molecular symmetry, $N_2^*$ exhibits high stability and lives for up to 1.3 s[27], which allows its involvement in chemical reactions, such as the above-mentioned $N_2$ disproportionation. When the disproportionation of $N_2$ is carried in water plasma (Fig. 1a) singlet $N_2$ could be easily activated to triplet $N_2^*$ through collisions with high-energy $(H_2O)_2^{+\bullet}$ and other species in water plasma. When the reaction between $N_2$ and $(H_2O)_2^{+\bullet}$ is carried inside the ion trap, singlet $N_2$ could be activated to triplet $N_2^*$ through collisions with $(H_2O)_2^{+\bullet}$ species activated by RF field (Fig. 1c). Accordingly, no reaction products were observed when $(H_2O)_2^{+\bullet}$ ions were trapped in $N_2$ gas without activation (Supplementary Fig. 1i). Also, note that, being an electronic transition, $N_2$ activation to $N_2^*$ occurs on a much shorter time scale compared to atomic rearrangements. Therefore, the event of $N_2$ activation to $N_2^*$ and the following association between $N_2^*$ and $(H_2O)_2^{+\bullet}$ could occur within a single collision between $N_2$ and $(H_2O)_2^{+\bullet}$.

In agreement with previous theoretical and experimental reports, our calculations indicate the co-existence of two $(H_2O)_2^{+\bullet}$ configurations, i.e., hydrogen-bonded $[H_3O^+\bullet\bullet\bullet OH]$ and $[H_2O\bullet\bullet\bullet OH_2]^+$ (+0.3 eV)[17]. Despite the $[H_3O^+\bullet\bullet\bullet OH]$ configuration being the global energy minimum for $(H_2O)_2^{+\bullet}$, we could not find a stable intermediate structure for the binding of $N_2^*$ with $[H_3O^+\bullet\bullet\bullet OH]$. In contrast, we could easily locate a stable intermediate structure for the binding of $N_2^*$ with the $[H_2O\bullet\bullet\bullet OH_2]^+$ configuration (Fig. 2b). The association of $[H_2O\bullet\bullet\bullet OH_2]^+$ configuration with $N_2^*$ occurs due to stabilization of the positive charge jointly by the $N_2$ and the two $H_2O$ moieties (INT1*, Fig. 2b). The INT1* structure is then converted into the excited-state intermediate HONH·HNOH$^{+\bullet*}$ (INT3*, $\Delta E = -1.4$ eV, Fig. 2b) by the direct double-proton transfer through an excited transition state (TS*) structure (Fig. 2b, Supplementary Fig. 3). The HONH·HNOH$^{+\bullet*}$ intermediate spontaneously and irreversibly dissociates into neutral HNO and cationic $NH_2OH^{+\bullet}$ ($\Delta E = -1.6$ eV, Fig. 2b). In addition to the dissociation channel, HONH·HNOH$^{+\bullet*}$ could also relax to its ground state, HONH·HNOH$^{+\bullet}$ (INT3, $\Delta E = -2.8$ eV, Fig. 2b), without dissociation. It is probably through this latter channel that the stable HONH·HNOH$^{+\bullet}$ signal is detected in our experiments (*m/z* 64, Fig. 1b). The HONH·HNOH$^{+\bullet}$ structure could dissociate into HNO and $NH_2OH^{+\bullet}$ by collisional activation ($\Delta E = 1.2$ eV, Fig. 2b), just as observed in the ion trap experiments (Fig. 1d). The $NH_2OH^{+\bullet}$ product can be neutralized to $NH_2OH$ by electron transfer from the environment.

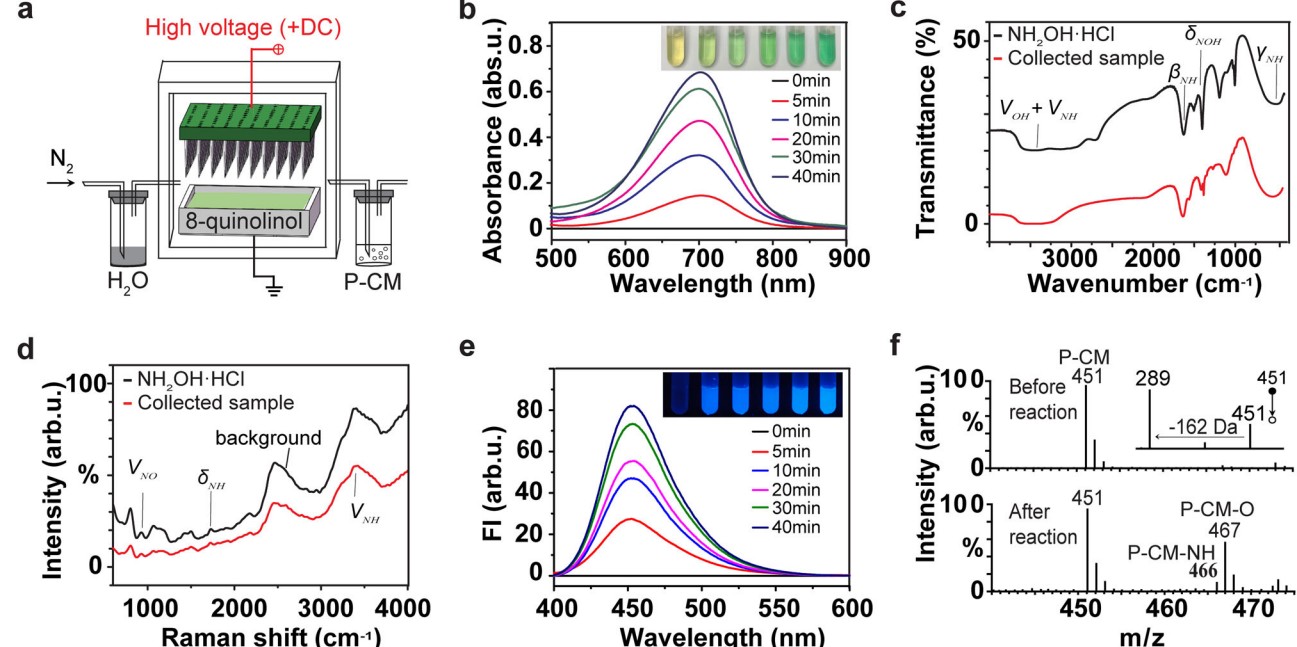

**Fig. 3 | Products of ambient disproportionation reaction of $N_2$ with $(H_2O)_2^{+\bullet}$ characterized by spectral methods. a** Schematic illustration of the reaction assembly for scale-up reaction and the collection of reaction products. DC: direct current. The 76 anodes of the array were connected to the same positive terminal of the DC high voltage power. The figure is adapted with permission from refs. [18],[20],[21]. **b** Ultraviolet-visible spectra of indooxine formed through the online reaction of the collected $NH_2OH$ with 8-quinolinol probe at different times of the reaction between $N_2$ and $(H_2O)_2^{+\bullet}$. **c** Infrared spectra of the collected sample (red) and $NH_2OH \cdot HCl$ standard (black). **d** Raman spectra of the collected sample (red) and $NH_2OH \cdot HCl$ standard (black). **e** Fluorescence spectra of 7-hydroxycoumarin formed through the online reaction of the collected HNO with P-CM probe at different times of the reaction between $N_2$ and $(H_2O)_2^{+\bullet}$. **f** Mass spectra of P-CM solution before and after collection of the reaction mixture, showing the formation of P-CM aza-ylide (P-CM-NH) and P-CM oxide (P-CM-O) due to the reaction between P-CM and HNO. The inset figure shows the tandem mass spectrum of protonated P-CM at *m/z* 451. P-CM: coumarin-based fluorescent probe.

Overall, it is clear that $N_2$ fixation by $(H_2O)_2^{+\bullet}$ via $HONH \cdot HNOH^{+\bullet}$ intermediate is mechanistically distinct from the earlier described processes of catalytic nitrogen fixation by molecular hydrogen at a heterogeneous surface[28], in which nitrogen reduction on metal catalyst usually proceeds via $N_2H_x$-type intermediates ($0 \leq x \leq 2$). Therefore, this report presents a peculiar mechanism of $N_2$ fixation.

## Scale-up reaction

The disproportionation reaction of $N_2$ with $(H_2O)_2^{+\bullet}$ was scaled up under ambient conditions using the setup in Fig. 3a, which consisted of an array of 76 needles evenly distributed on a ~ $3.5 \times 5.5$ cm² tungsten plate to generate abundant $(H_2O)_2^{+\bullet}$ and the accessories to collect the $NH_2OH$ and HNO products. We found that $NH_2OH$ was most efficiently collected at the cathode electrode, while HNO was most efficiently collected through the neutral outlet line (Fig. 3a). These observations further support the conclusion that $NH_2OH$ in the reaction was formed in cationic form while HNO was formed in neutral form. Also, these observations enabled special experimental design whereby the HNO and $NH_2OH^{+\bullet}$ products could be spatially separated upon collection: the positively charged $NH_2OH^{+\bullet}$ product is driven by electric field into collection plate at the cathode, while the neutral HNO product is carried with the $N_2$ stream to the bottle at the upper outlet of the reactor (Fig. 3a). This experimental design effectively prevents the back reaction between HNO with $NH_2OH$ to form $N_2$ and $2H_2O$. Note that according to our calculations the back reaction between the HNO and $NH_2OH$ products while in the gas phase (e.g., in situ near the discharge area) is hindered by the rather significant energy barrier of ca. 1.4 eV.

Under the optimized experimental conditions, the production of 18.5 μM $NH_2OH$ and 17.7 μM HNO products could be achieved within just 10 min, as quantified by standard spectrophotometric methods (Supplementary Fig. 4, Supplementary Fig. 5). The collected $NH_2OH$ was reacted with an optical probe 8-quinolinol to form indooxine,

which was quantified by ultraviolet-visible (UV-Vis) spectroscopy[29]. The signal intensity showed clear correlation with the time of the reaction between $N_2$ and $(H_2O)_2^{+\bullet}$ (Fig. 3b). The formation of indooxine due to the reaction between $NH_2OH$ and 8-quinolinol was further confirmed by tandem MS experiments through the comparison with indooxine standard (Supplementary Fig. 6a). The formation of $NH_2OH$ product due to the disproportionation reaction between $N_2$ and $(H_2O)_2^{+\bullet}$ was further validated by infrared and Raman (Fig. 3c, d) spectroscopy of the collected samples.

Similarly, the collected HNO was reacted with a coumarin-based fluorescent probe, called P-CM, to form 7-hydroxycoumarin. The 7-hydroxycoumarin was quantified by fluorescence spectroscopy. The signal intensity showed clear correlation with the time of the reaction between $N_2$ and $(H_2O)_2^{+\bullet}$ (Fig. 3e). The occurrence of the reaction between P-CM and the HNO product was further confirmed by the detection of other products of the reaction between P-CM and HNO: the P-CM aza-ylide and P-CM oxide (Fig. 3f, Supplementary Fig. 6b), which were in good agreement with the literature data[30]. Note that the preparation and characterization of P-CM is described in the Supplementary Fig. 6c, d.

A series of reference experiments further confirmed the identify of $NH_2OH$ product and its formation by $N_2$ reduction. No $NH_2OH$ signal was detected either in the blank control setup or in the Ar atmosphere ($N_2$ was replaced by Ar in Fig. 3a), while clear $NH_2OH$ signal was detected in both the synthetic air atmosphere and pure $N_2$ atmosphere (Supplementary Fig. 7). The efficiency of $NH_2OH$ product formation in synthetic air was about 50% of that in $N_2$, which may be due to the lower content of $N_2$ in the synthetic air as well as due to the influence of $O_2$. Further, the proton nuclear magnetic resonance (¹H NMR) spectrum of the condensate collected in the reaction between $N_2$ and water plasma displayed a peak at 10 ppm, which matched the peak generated by standard $NH_2OH \cdot HCl$ (Supplementary Fig. 8a). Consistently, the

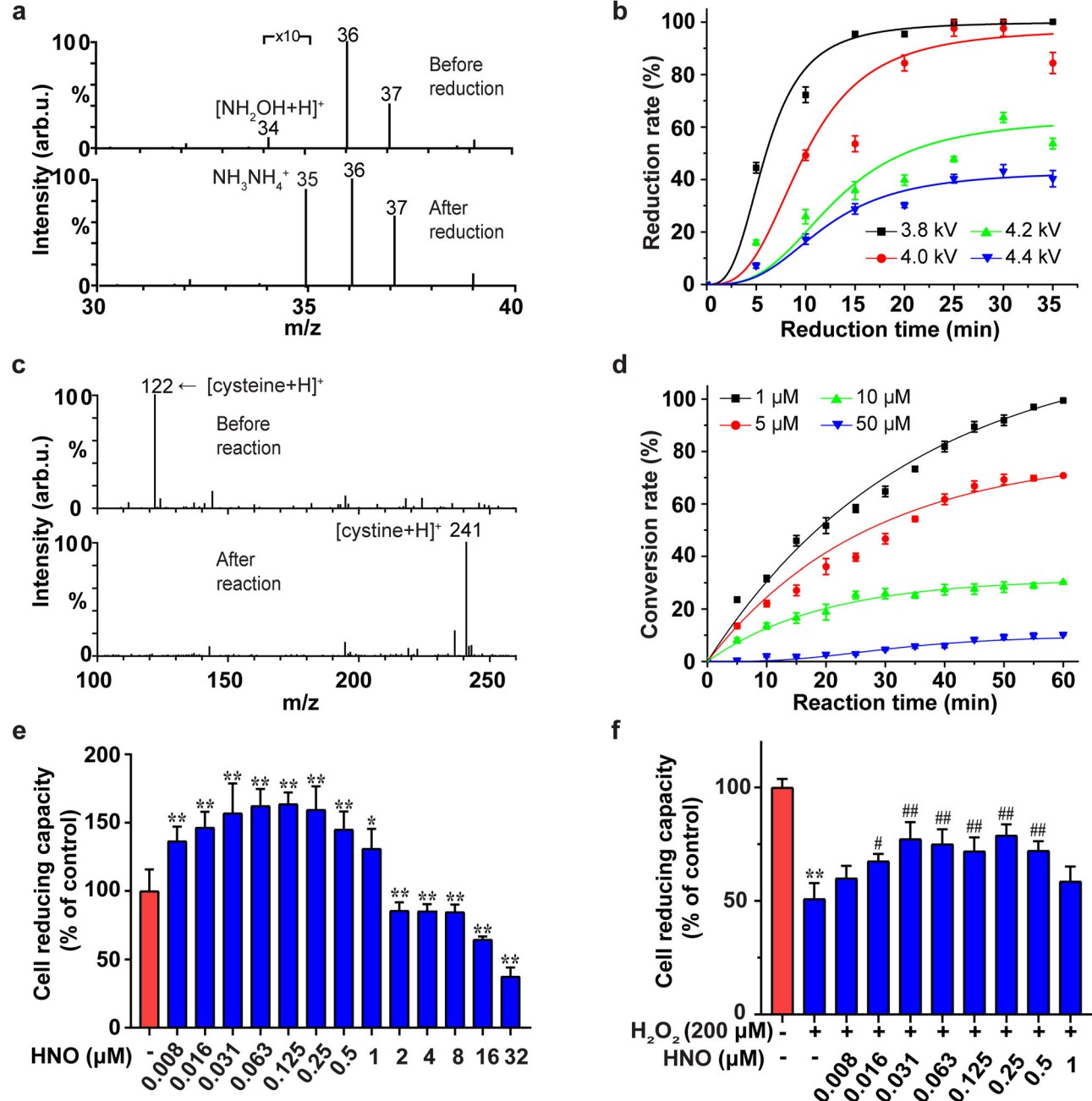

**Fig. 4 | Further validation and application of the NH₂OH and HNO products of N₂ disproportionation with (H₂O)₂⁺˙.** **a** Electrolytic reduction of the collected NH₂OH product (corresponding signal at $m/z$ 34) into NH₃ (corresponding signal at $m/z$ 35) confirmed by mass spectrometry detection. The signals at $m/z$ 36 and $m/z$ 37 correspond to $(H_2O)_2^{+\cdot}$ and $H^+(H_2O)_2$. **b** The kinetics of electrolytic reduction of NH₂OH collected at different discharge voltages into NH₃ determined using the indophenol blue method[42] (see Supplementary Fig. 9). **c** The conversion of cysteine (corresponding signal at $m/z$ 122) into cystine (corresponding signal at $m/z$ 241) via the reaction of cysteine with the collected HNO product of the disproportionation reaction of N₂ confirmed by mass spectrometry detection. **d** The kinetics of cysteine conversion into cystine via the reaction with the collected HNO product (~25 μM; collection time 30 min) at different cysteine concentrations. **e** Effects of different concentration levels of HNO alone on HT22 cell reducing capacity. **f** HT22 cells were pretreated with different concentration levels of HNO for 24 h and incubated with or without H₂O₂ (200 μM) for 1 h. Cell reducing capacity as determined with Cell Counting Kit-8 assay. $^*P < 0.05$ and $^{**}P < 0.01$ versus control, $^{\#}P < 0.05$ and $^{\#\#}P < 0.01$ versus model. The error bars indicate the standard deviation ($n = 3$).

experiments with $^{15}N_2$ instead of $^{14}N_2$ displayed the characteristic $^{15}NH_2$ peak (Supplementary Fig. 8a).

## Application of the reaction products

Note that both NH₂OH and HNO products of the disproportionation reaction between N₂ and (H₂O)₂⁺˙ are highly valuable chemicals. Further, using a low-cost homemade setup powered by a 1.5 V solar cell battery (Supplementary Fig. 9) in our laboratory, we were able to easily reduce the prepared NH₂OH into NH₃ (Fig. 4a, Supplementary Fig. 9) with the conversion rate of almost 100% within ca. 30 min (Fig. 4b). The simplicity of the experimental assembly suggests that the disproportionation reaction between N₂ and (H₂O)₂⁺˙ could probably serve the agricultural fields as tiny onsite ammonia plants, which could be powered by solar cells[31]. We also showed that the HNO product could be used to directly convert cysteine into cystine (Fig. 4c) with the conversion rate of almost 100% within ca. 60 min (Fig. 4d) under the

conditions tested. This observation suggests that HNO could be used for the specific chemical modification of thiols in proteins.

In model experiments, we demonstrated the potential utility of the collected HNO product in promoting the proliferation of HT22 cells and protecting them from $H_2O_2$-induced oxidative stress (Fig. 4e, f, Supplementary Fig. 10, Supplementary Fig. 11). While the low dose of HNO (0.008–1 μM) remarkably increased cell reducing capacity, as measured by Cell Counting Kit-8 (CCK-8), a high dose of HNO (1–32 μM) was found to inhibit cell reducing capacity (Fig. 4e, Supplementary Fig. 10). These observations suggest an important role of HNO in regulating cell growth[32]. It has been reported that HNO can interact with and modify several protein targets, such as thiol proteins, stimulating cell proliferation[33]. In addition to that, through the reaction with the dehydrogenase and tetrazolium salts, HNO could also convert to NO, which further regulates cell growth. Indeed, NO has been established as a potent modulator of cell proliferation and the cell cycle, including stimulatory and inhibitory effects[34,35]. We propose that the observed two-phased effect of HNO on cell reducing capacity could be attributed to a complex signaling pathway involving NO. Further investigation is required to clarify the effects of HNO on cell reducing capacity.

## Characteristics of the $N_2$ disproportionation reaction

Using the total plate area (~ 3.5 × 5.5 cm²) as the active area of the 76-pin device (Fig. 3a), we estimated the Faradic efficiency (FE) ≈ 64% with the yield ≈ 1.14 μg cm$^{-2}$ h$^{-1}$ for $NH_2OH$ and the FE ≈ 20% with the yield ≈ 0.37 μg cm$^{-2}$ h$^{-1}$ for HNO (optimized in Supplementary Fig. 12). Note that the maximum yields calculated based only on the total effective surface area of 76 needle tips (as the plasma is activated only at the tip surface of the needles) were $5.10 \times 10^3$ μg cm$^{-2}$ h$^{-1}$ for $NH_2OH$ and $1.53 \times 10^3$ μg cm$^{-2}$ h$^{-1}$ for HNO, respectively (see details in Supplementary Note 2). The lower yield and FE for HNO is probably caused by its higher reactivity, resulting in the partial conversion of HNO by $H_2O$ and other chemicals in the collected solution. Thus, in MS (Supplementary Fig. 13a-d), ion chromatography (Supplementary Fig. 13e-f), and UV-Vis absorption spectroscopy (Supplementary Fig. 14) we also observed $NO_3^-$, $NO_2^-$ and $H_2O_2$ products, which were likely produced by further conversion of the HNO product. The content of $NO_3^-$, $NO_2^-$ and $H_2O_2$ produced over 10 min was estimated to be about 0.007 mM, 0.004 mM, and 0.03 mM, respectively. Interestingly, ion chromatography (Supplementary Fig. 15) and $^1$H NMR spectroscopy (Supplementary Fig. 7b) data also indicated the production of $NH_4^+$ (about 0.02 mM over 10 min), which is likely through the reduction of $NH_2OH$ product. The simple reduction of $NH_2OH$ to $NH_3$ is demonstrated by the results in Fig. 4a. The origin of $NH_3$ via the reduction of $N_2$ by water plasma was further confirmed by isotope-labeling experiments (Supplementary Fig. 7b).

By integrating the calculated yields of N-containing products, the conversion rate of $N_2$ under the optimum conditions was estimated as ca. 0.001%, which is higher than in other plasma methods (for details see Supplementary Note 3)[36]. We believe that the higher conversion rate in our method is mainly due to the formation of abundant $(H_2O)_2^{+\bullet}$ cations, which act as efficient activators of $N_2$ molecule. As $NH_2OH$ can be converted to $NH_3$ with nearly 100% efficiency, the estimated $NH_3$ production rate under the optimum conditions was ca. 1.8 μg cm$^{-2}$ h$^{-1}$ (for details see Supplementary Note 3).

In comparison with the HB process and other methods of $N_2$ fixation, including catalytic and plasma methods (Tables 1 and 2), our method offers considerably higher economy efficiency (1.54 \$ kwh$^{-1}$ for $NH_2OH$ and 7310 \$ kwh$^{-1}$ for HNO), which is owing to the high value of $NH_2OH$ and particularly HNO products as compared to $NH_3$. In terms of energy cost (g kwh$^{-1}$), our method is currently ca. 3 orders of magnitude less efficient than HB and ca. one to two orders of magnitude less efficient than plasma-based methods for $N_2$ fixation reported

earlier (0.154 g kwh$^{-1}$ for $NH_2OH$ and 0.149 g kwh$^{-1}$ for HNO). The higher energy cost in our method is mainly due to the avoidance of a catalyst, high pressures and high temperatures. The energy cost is expected to reduce as the plant continues to be upgraded. In comparison with several catalytic methods for $N_2$ fixation under mild conditions (near room temperature and atmospheric pressure) reported earlier, our method offers similar yield of N-containing products (1.14 μg·cm$^{-2}$·h$^{-1}$ for $NH_2OH$ and 0.37 μg·cm$^{-2}$·h$^{-1}$ for HNO). The product yield of $NH_2OH$ was ca. two times higher than that of $NH_3$, while the product yield of HNO was ca. three times lower than the integral product yield of $NO_3^-$ and $NO_2^-$ (Table 1). The lower yield of HNO compared to $NO_3^-$ and $NO_2^-$ probably reflects the high reactivity of HNO intermediate to give $NO_3^-$ and $NO_2^-$. It is a challenge for further research to optimize the reaction parameters toward the higher yield and stability of the high-value HNO product.

Overall, key merits of our method include: (1) mild conditions; low cost; easy implementation; scalability; (2) high-value products: according to the current market, the potential value about 1.5 \$ of $NH_2OH$ and 7310 \$ of HNO were produced by 1 kWh of electricity (≤0.2 \$); (3) no catalyst needed; (4) high atom economy: the oxidation of one nitrogen atom of $N_2$ to HNO coupled with the simultaneous reduction of the other nitrogen atom of $N_2$ to $NH_2OH$.

In summary, we have demonstrated that the atmospheric $N_2$ can be disproportionately fixed by $(H_2O)_2^{+\bullet}$ under ambient conditions into economically valuable $NH_2OH$ and HNO, presenting an alternative to the current necessity of fixing $N_2$ into $NH_3$. The combination of the essential N, O and H atoms in the obtained products considerably increases variability of possible chemistries as compared to $NH_3$. The experimental and theoretical studies indicate that triplet-state $N_2$ is activated by $(H_2O)_2^{+\bullet}$ to form intermediate HONH·HNOH$^{+\bullet}$, which is further decomposed to form $NH_2OH^{+\bullet}$ and HNO. The mechanism of $N_2$ fixation by the 2c-3e $(H_2O)_2^{+\bullet}$ structure through the excited-state double-proton transfer is principally different from the previously proposed methods. Remarkably, the formation of $NH_2OH$ and HNO product occurs in the gas phase, which opens potentialities for their direct in-situ application, without the need of sample collection. The design ideas in this work could motivate more research efforts to further explore the potential of distinct $(H_2O)_2^{+\bullet}$ chemistry and open alternative possibilities for green nitrogen fixation.

## Methods

### Chemicals and material

NH₂OH·HCl, 8-quinolinol and 2-(diphenylphosphino) benzoic acid were purchased from Shanghai Sun Chemical Technology Co., Ltd (China), with a purity >99%. $D_2O$ was purchased from Cambridge Isotope Laboratories, Inc. (Andover, MA, USA), with a purity >99%. Authentic Angeli's salt (AS) used in fluorescence quantification experiments was purchased from Cayman Chemical and stored at −20 °C, with a purity >99%. Ultra-purity $N_2$ (>99.999%), ultra-purity helium (>99.999%), ultra-purity argon (>99.999%) and ultra-purity $^{15}N_2$ (>99%) were obtained from Jiangxi Guoteng Gas Co. Ltd (Nanchang, China). Water used in all experiments was purified by a Milli-Q system (Millipore, USA).

Foetal bovine serum (FBS) was purchased from Zhejiang Sorfa Life Science Co., Ltd. (Zhejiang, China). Dulbecco's modified eagle's medium (DMEM), Trypsin-ethylene diamine tetraacetic acid solution and phosphate buffered saline (PBS) were obtained from Solarbio Life Science Co., Ltd. (Beijing, China). $H_2O_2$ (30%) was purchased from Xilong Scientific Co., Ltd. (Guangdong, China). The CCK-8 assay kit was obtained from Biosharp Life Sciences Co., Ltd. (Anhui, China). Microplate reader was purchased from Gene Company Limited (Hong Kong SAR, China). $CO_2$ incubator was obtained from SANYO (Osaka, Japan). Inverted microscope was purchased from Leica (Germany). Superclean bench was from Suzhou Purification Co., Ltd. (Shanghai, China).

**Table 1 | Comparison with catalytic methods of nitrogen fixation under mild conditions**

| Method | Catalyst | Products | Conditions | Product yield ($\mu g \cdot cm^{-2} \cdot h^{-1}$) | Economy efficiency[a] $ cm^{-2} \cdot h^{-1}$ | FE (%) | Ref |
|---|---|---|---|---|---|---|---|
| Electrocatalytic | $MoS_2$ | $NH_3$ | RT[b], AP[c] | 5.39 | $0.3234 \times 10^{-6}$ | 1.17 | 44 |
| Electrocatalytic | PEBCD/C | $NH_3$ | RT, AP | 1.58 | $0.0948 \times 10^{-6}$ | 2.85 | 45 |
| Electrocatalytic | Au nanorods | $NH_3$ | RT, AP | 1.65 | $0.099 \times 10^{-6}$ | 3.88 | 46 |
| Electrocatalytic | Ru/C | $NH_3$ | 20 °C, AP | 0.21 | $0.0126 \times 10^{-6}$ | 0.28 | 47 |
| Electrocatalytic | Pt/C | $NH_3$ | RT, AP | 69.77 | $4.1862 \times 10^{-6}$ | 0.52 | 48 |
| Electrocatalytic | $Fe_2O_3$/CNTs | $NH_3$ | 20 °C, AP | 0.22 | $0.0132 \times 10^{-6}$ | 0.15 | 49 |
| Electrocatalytic | AuHNCs | $NH_3$ | RT, AP | 3.90 | $0.234 \times 10^{-6}$ | 30.2 | 50 |
| Electrocatalytic | B-graphene | $NH_3$ | RT, AP | 9.80 | $0.588 \times 10^{-6}$ | 10.8 | 51 |
| Photocatalytic | P3MeT/$TiO_2$ | $NH_3$ | 20 °C | 0.03 | $0.0018 \times 10^{-6}$ | N/A | 52 |
| Photocatalytic | Au/black Si/Cr | $NH_3$ | N/A | 1.33 | $0.0798 \times 10^{-6}$ | N/A | 53 |
| Photocatalytic | $TiO_2$/Au/a-$TiO_2$ | $NH_3$ | RT | 0.23 | $0.0138 \times 10^{-6}$ | N/A | 54 |
| Photocatalytic | Au NPs/Nb-$SrTiO_3$/Zr/ZrOx | $NH_3$ | RT | 0.01 | $0.0006 \times 10^{-6}$ | N/A | 55 |
| Biological | Enzyme | $NH_3$ | Antarctica | $5.9 \times 10^{-3}$ | N/A | N/A | 56 |
| Disproportionation by $(H_2O)_2^{+\cdot}$ | No | | RT, AP | 1.14[d] | $1.14 \times 10^{-5}$ | | This work |
| | | $NH_2OH$ | | $(5.10 \times 10^3)$[e] | 0.051 | 64.0 | |
| | | HNO | | 0.37[d] | 0.0555 | 18.0 | |
| | | | | $(1.53 \times 10^3)$[e] | 229.5 | | |
| | | $NH_3$ | | 0.63[d] | $6.3 \times 10^{-6}$ | 55.0 | |
| | | $NO_2^-$ | | 0.34[d] | $0.07 \times 10^{-6}$ | 33.0 | |
| | | $NO_3^-$ | | 0.81[d] | $0.17 \times 10^{-6}$ | 96.0 | |

[a] Note that the economy efficiency for the catalytic methods does not account for the price of the catalyst. [b] RT: room temperature. [c] AP: atmospheric pressure.
[d] The yield$_{min}$, which is the yield obtained when the total surface area of the discharge needle array was accounted for. [e] The yield$_{max}$, which is the yield obtained when only the total effective surface area of the 76 needle tips was accounted for. In this work, the yield was given for $NH_2OH$ product. The $NH_2OH$ product could be converted into $NH_3$ with ca. 100% efficiency (Fig. 4b). N/A: Information not available in the article. For regarding the calculation of product yields see Supplementary Note 2.

**Table 2 | Comparison with plasma methods of nitrogen fixation**

| Approach | Catalyst | Products | Energy cost MJ $mol^{-1}$ | Energy yield g $kwh^{-1}$ | Economy efficiency $ $kwh^{-1}$ | Ref |
|---|---|---|---|---|---|---|
| [a]DBD plasma | Copper wire | $NH_3$ | 18.5 | 3.308 | 0.1985 | 57 |
| DBD plasma | Alumina medium | $NH_3$ | 33.4 | 1.832 | 0.1099 | 58 |
| DBD plasma | ferroelectric materials | $NH_3$ | 68 | 0.900 | 0.0540 | 59 |
| DBD plasma | Ru/Mg/$Al_2O_3$ | $NH_3$ | 1.71 | 35.8 | 2.1474 | 60 |
| DBD plasma | Ru/$Al_2O_3$ | $NH_3$ | 32.4 | 1.9 | 0.1133 | 61 |
| Non-thermal plasma | Ru/Cs/carbon nanotubes | $NH_3$ | 26.6 | 2.3 | 0.1380 | 62 |
| Nonequilibrium plasma | No | $NH_3$ | 95 | 0.644 | 0.0387 | 36 |
| Plasma-water droplet | No | $NH_3$ | 9500 | 0.006 | 0.0004 | 63 |
| Plasma-activated nitrogen fixation | No | $NH_3$ | 0.61 | 100.3 | 6.0197 | 10 |
| [b]UV irradiation plasma | No | $NH_3$ | 988 | 0.062 | 0.0037 | 64 |
| Plasma electrolytic system | No | $NH_3$ | 139 | 0.440 | 0.0264 | 65 |
| [c]DC plasma-driven electrolysis | No | $NH_3$ | 100 | 0.612 | 0.0367 | 66 |
| | | $NO_3^-$ | | 2.232 | 0.0134 | |
| DBD plasma | No | $NO_3^-$ | 8 | 27.90 | 0.1674 | 67 |
| DBD plasma | No | $NO_2^-$ | 1175 | 0.141 | 0.0008 | 68 |
| Disproportionation by $(H_2O)_2^{+\cdot}$ | No | $NH_2OH$ | 770 | 0.154 | 1.5429 | This work |
| | | HNO | 2290 | 0.049 | 7310.0 | |
| | | $NH_3$ | 504 | 0.121 | 0.007 | |
| | | $NO_2^-$ | 2535 | 0.065 | 0.01 | |
| | | $NO_3^-$ | 1488 | 0.155 | 0.03 | |

[a] DBD: dielectric barrier discharge. [b] UV: ultraviolet. [c] DC: direct current. For Details regarding the calculation of energy cost see Supplementary Note 4.

A coumarin-based fluorescent probe, P-CM, was synthesized in our lab according to the method reported by Tan et al with modification, with a purity >99%[30]. In detail, 2-(diphenylphosphino) benzoic acid (306 mg, 1 mmol) was dissolved in 50 mL of anhydrous $CH_2Cl_2$ under inert atmosphere. Then, 4-(dimethylamino) pyridine (6.1 mg, 0.05 mmol) and 1-ethyl-3-(3-dimethylaminopropyl) carbodiimide hydrochloride (191.7 mg, 1 mmol) were added, and the reaction mixture was stirred at room temperature for 30 min. 7-Hydroxycoumarin (194.6 mg, 1.2 mmol) was then added, and the resulting mixture was stirred at room temperature. The reaction mixture was concentrated under reduced pressure. The residue was purified by the silica gel chromatography (ethyl acetate/ petroleum ether = 1:2, v/v) to yield P-CM product as a faint yellow solid (423 mg, 0.93 mmol). The isolated mass and yield were calculated to be ~423 mg and 93%, respectively. Even though the reaction was conducted at room temperature, it is essential to note that some reagents, such as petroleum ether, are highly volatile. Hence, in order to ensure the safety of experiments, appropriate control safeguards are needed to prevent potential hazards. $^1H$ NMR ($CDCl_3$, 500 MHz) δ(ppm): 8.278–8.245 (m, 1H), 7.681–7.657 (d, J = 9.6 Hz, 1H), 7.502–7.474 (m, 2H), 7.440–7.418 (m, 1H), 7.377–7.277 (m, 10H), 7.022–6.989 (m, 1H), 6.930–6.902 (m, 2H), 6.396–6.372 (d, J = 9.6 Hz, 1H). MS (ESI) $m/z$ 451.0 $[M + H]^+$ (Fig. 3f).

### Reaction between $N_2$ and $(H_2O)_2^{+\bullet}$ with real-time MS

Experimental setup to study the products of the reaction between $N_2$ and $(H_2O)_2^{+\bullet}$ in real time is shown in Fig. 1a. In order to produce $(H_2O)_2^{+\bullet}$, neutral Ar gas was bubbled through liquid $H_2O$ at a flow rate of 10 - 100 mL min$^{-1}$, and the produced $H_2O$/Ar vapor was guided to the home-made ambient corona discharge ion source through Teflon tubing with inner diameter (ID) of 0.75 mm. To build the ion source, a sharp stainless-steel needle (outer diameter (OD) 150 μm; end curvature radius ~30 μm) was used as the discharge electrode for plasma. The needle was coaxially inserted into a fused silica capillary (ID 0.25 mm, OD 0.40 mm) and was fixed coaxially with a union tee and silica ferrule. The back end of the tee was connected to the $H_2O$/Ar line. The distance from the inlet of the mass spectrometer capillary to the tip of the ion source was 20 mm. The direct current (DC) voltage of +2.0 kV was applied to the stainless-steel needle in order to generate ambient corona discharge. A flow of neutral $N_2$ (100 mL min$^{-1}$) was introduced through a separate channel to interact with the produced $(H_2O)_2^{+\bullet}$. In a reference experiment, $N_2$ was replaced by Ar (Supplementary Fig. 1a).

### Scale-up reaction between $N_2$ and $(H_2O)_2^{+\bullet}$

The experimental setup for the scaled-up disproportionation reaction between $N_2$ and $(H_2O)_2^{+\bullet}$ with the collection of reaction products is presented in Fig. 3a. High-purity $N_2$ (99.999 %) controlled by mass flow controller was bubbled through liquid $H_2O$ and was then transferred through the discharge reactor in quartz enclosure toward the outlet tubing connected with a collection bottle. The reactor consisted of a discharge array of 76 tungsten needles under DC voltage (anode) and a flat grounded electrode (cathode). The distance between the tips of two adjacent needles was evenly set as 0.5 cm. The 76 anodes of the array were connected to the same positive terminal of the DC high voltage power. Cationic reaction product $NH_2OH^{+\bullet}$ was collected by an indium tin oxide (ITO) coated glass bottle filled with 8-quinolinol probe solution (6 mL), which was placed on the top of the cathode. Neutral reaction product HNO was pumped out through the outlet line into the bottle filled with P-CM probe solution (2 mL) (Fig. 3a). The amount of electric charge neutralized on the cathode plate was determined by an electrochemical workstation (Shanghai Chenhua Instrument Co., Lt, CHI660E).

### MS settings

MS detection was carried out using an LTQ-XL ion trap mass spectrometer (LTQ-XL, Thermo Scientific, San Jose, CA, USA). The temperature of the ion transfer capillary was 150 °C. The capillary voltage was 1.0 V. The tube lens voltage was 30.0 V. The pressure of ion trap was $1 \times 10^{-5}$ torr. High-purity helium (99.999%) was used as the collision gas. The CID-MS experiments were performed by applying excitation alternating current voltage to the end caps of the ion trap to induce collisions of the isolated ions. The CID-MS spectra were obtained by activation of the precursor ions at the normalized collision energy varied from 0% to 50%. Ion detection was done in the positive ion mode. Other LTQ-XL parameters were automatically optimized by the system.

### Quantification of $NH_2OH$ by UV-Vis spectroscopy

The concentration of collected $NH_2OH$ was determined by the 8-quinolinol color test method with modification[29]. In detail, 1.2 mL phosphate buffer (0.05 M, pH 6.8), 0.24 mL trichloroacetic acid solution and 1.2 mL sodium carbonate solution (1.0 M) were mixed in a tube. Then 1.2 mL ethanol solution of 8-quinolinol (10 mg mL$^{-1}$) was added. Finally, pure water was added into the tube to bring the solution volume to 6 mL. Thus prepared 6 mL reagent solution was placed on top of the cathode plate. The $N_2$ disproportionation reaction was then started. The UV-Vis absorption spectra of the solution were measured at different reaction times (0, 5, 10, 20, 30, 40 min) on a spectrophotometer (UV-1900i, Shimadzu, Japan). The formation of indooxine due to the reaction between $NH_2OH$ and 8-quinolinol was evident by the characteristic absorbance band with the maximum at 705 nm. For calibration, UV-Vis spectra were collected for a series of blank (zero time of disproportionation reaction) reagent solutions spiked with different concentrations of $NH_2OH$ (0.01 mM, 0.05 mM, 0.1 mM, 0.2 mM, 0.25 mM), showing good linear correlation of indooxine absorbance with $NH_2OH$ concentration by three independent measurements (Supplementary Fig. 5a, b). Blank UV-Vis measurements were also done when no $NH_2OH$ was added.

### Infrared and Raman spectroscopy

The collected samples and $NH_2OH \cdot HCl$ standard were analyzed on a Thermo Nicolet iS5 Fourier transform infrared spectrograph. In detail, KBr samples were powdered and spread evenly on top of the cathode plate (Fig. 3a). The $N_2$ disproportionation reaction was then run over 24 h. The sample collected on the cathode plate was analyzed by infrared spectroscopy. $NH_2OH \cdot HCl$ sample was measured for control.

The collected samples and $NH_2OH \cdot HCl$ standard were also analyzed by Via RM2000 laser Raman spectroscopy (Renishaw company, United Kingdom) with 10 s integration time in the spectral range of 400–4000 cm$^{-1}$. Continuous-Wave laser irradiation at the wavelength of 532 nm was focused through a 50X objective to excite the samples. Deionized water was placed on top of the cathode plate (Fig. 3a). The $N_2$ disproportionation reaction was run over 48 h. The product collected on the cathode plate was analyzed by Raman spectroscopy. The authentic $NH_2OH \cdot HCl$ solution was analyzed for control.

### Quantification of HNO by fluorescence spectroscopy

The amount of collected HNO was estimated by the method reported by Tan et al[30]. P-CM probe was freshly dissolved in $N,N$-dimethylformamide to obtain 1 mM stock solution, which was diluted to 10 μM with 100 mM PBS buffer (pH 7.4). The prepared P-CM solutions (10 μM, total volume = 2 mL) were connected to the reactor outlet line as shown in Fig. 3a for different periods of reaction time at room temperature. The fluorescence of 7-hydroxycoumarin formed through the reaction of HNO with the P-CM probe was excited at 370 nm and was measured at 450 nm. Fluorescence from the blank P-CM (10 μM) solution (not reacted with HNO) was measured as control. The yields of HNO were calculated from a standard curve using AS as HNO source. A stock solution of AS (2 μM mL$^{-1}$) was prepared in 1 M NaOH solution and stored at −20 °C. 20 μL AS standard solutions with different concentrations (0.01 μM, 0.25 μM, 1 μM, 2 μM, 5 μM) were separately

added into 10 μM P-CM solutions to bring the volume to 2 mL. Then, fluorescence spectra were recorded. The fitting curve showed good linearity of fluorescence intensity at 450 nm with HNO concentration by three independent measurements (Supplementary Fig. 5c, d).

Noteworthy, while HNO is known to dimerize at high concentrations[37], the collected HNO solutions displayed considerable stability over the period of at least 2 days (Supplementary Fig. 11). The stability of HNO product is due to its relatively low concentration (sub mM), which prevents significant dimerization of HNO in the collected solution. Our observations are consistent with the recent reports indicating long-term stability of HNO in aqueous solutions under physiologically relevant conditions[38,39].

## Ion chromatography
The amounts of $NH_3$, $NO_3^-$ and $NO_2^-$ formed over the interaction between water plasma and $N_2$ were also estimated by ion chromatography (Thermo-Fisher DIONEX ICS-1100). The test procedure followed the standard operation procedure provided by the vendor. Briefly, 25 μL of the resultant was injected into the chamber for separation. Delivery speed was 4 mL min$^{-1}$. The peak eluted at the retention time of 9 min was assigned to $NH_4^+$. The amount of $NH_4^+$ was estimated using a calibration curve by standard solutions (Supplementary Fig. 15). The peaks eluted at the retention time of 5 min and 8 min were assigned to $NO_2^-$ and $NO_3^-$. The amounts of $NO_2^-$ and $NO_3^-$ were estimated using a series of standard solutions (Supplementary Fig. 13e, f).

## Quantification of $H_2O_2$ by UV-Vis spectroscopy
The amount of $H_2O_2$ was measured by UV-Vis absorption spectroscopy (UV-1900i, Shimadzu, Japan). According to reference[40], potassium permanganate was reduced by hydrogen peroxide in sulfuric acid with the maximum absorption peak at 525 nm. The amount of $H_2O_2$ was derived from the amount of consumed $MnO_4^-$ for calibration with a series of standard solutions (Supplementary Fig. 14).

## Experiments on HT22 cells treated with HNO product
It was reported that $H_2O_2$ signaling via thiol modification may lead to radical-mediated cellular damage (i.e., lipid peroxidation, protein carbonyl formation, etc.) due to the Fenton reaction[33]. In contrast, HNO demonstrates the ability to provide neuroprotection, and inhibit deleterious radical chain reaction, oxidative stress and lipid peroxidation[41]. Thus, HNO could interact with thiols thereby reducing the toxic effect of $H_2O_2$. In order to explore the possibility of using HNO to protect cells against $H_2O_2$ toxicity, we incubated HT22 cells with HNO and then treated them with $H_2O_2$.

In the present experiments, we established $H_2O_2$-induced oxidative stress models in HT22 cells to investigate whether HNO has anti-oxidative stress and neuroprotective effects. HNO was generated using the disproportionation reaction of $N_2$ with $(H_2O)_2^{+•}$ under ambient conditions. HT22 cells were purchased from Procell Life Science & Technology Co., Ltd. and cultured in DMEM supplemented with 10% (v/v) FBS and 1% penicillin/streptomycin at 37 °C with the atmosphere of 5% $CO_2$. To induce cell damage, $H_2O_2$ was added to the culture for 1 h.

The HT22 cells were seeded in 96-well plates ($0.8 \times 10^5$ cells/well) and incubated with different concentrations of HNO for 24 h followed by $H_2O_2$ treatment. Cell reducing capacity was examined by the CCK-8 assay kit (Dojindo, Tokyo, Japan). The absorbance of the samples was measured at 450 nm by microplate reader. The cell survival ratio was expressed as the percentage of the control.

All of the statistical analyses were performed by the SPSS 21.0 software. All of the data are expressed as the mean ± SD. Comparison between the two groups was assessed with an unpaired $t$-test, while comparison among several groups was evaluated using one-way ANOVA. The $p$-value <0.05 was considered statistically significant.

## Calculation of the Faradaic efficiency and yield
For the experiment setup shown in Fig. 3a, the yields of $NH_2OH$ and HNO were calculated using the following equations reported in earlier literature[42]:

$$\text{Yield}_{NH2OH} = (C_{NH2OH} \times V_{NH2OH} \times M_{NH2OH})/(t \times S) \quad (1)$$

$$\text{Yield}_{HNO} = (C_{HNO} \times V_{HNO} \times M_{HNO})/(t \times S) \quad (2)$$

where $V$ is the volume of absorption solution ($V_{NH2OH} = 6$ mL; $V_{HNO} = 2$ mL), $t$ is the reaction time (10 min), $C$ is the concentration of the collected product over 10 min (e.g., at 4.2 kV: $C_{NH2OH} = 18.5$ μM and $C_{HNO} = 17.7$ μM, determined using calibration curves in Supplementary Fig. 5), $M$ is the molar mass of a collected product ($M_{NH2OH} = 33$ g mol$^{-1}$; $M_{HNO} = 31$ g mol$^{-1}$), $S$ is the total effective surface area to conduct the reaction. The $S$ value is measured by microscope (For details see Supplementary Note 2).

The FE for $N_2$ reduction was determined as the amount of electric charge used for the generation of $NH_2OH$ divided by the total charge passed through the cathode electrode during reaction. The FE for $N_2$ oxidation was determined as the amount of electric charge used for the generation of HNO divided by the total charge passed through the electrodes during reaction. Assuming that one electron is needed to produce one HNO molecule and one $NH_2OH$ molecule (Fig. 2a), the FE value was calculated according to the reference reported by Wang et al[42]:

$$FE = (n \times F \times C \times V)/Q \quad (3)$$

where $n$ is the number of electrons transferred in the reaction ($n = 1$ in our case); $F$ is the Faraday's constant ($F = 96485.33$ C mol$^{-1}$); $C$ is the concentration of the collected $NH_2OH$ or HNO; $V$ is the volume of absorption solution to collect $NH_2OH$ or HNO; and $Q$ denotes the total charge passed through the cathode electrode during reaction (0.0168 C over 10 min reaction at 4.2 kV).

## Reduction of $NH_2OH$ to $NH_3$
To quantify the amount of $NH_2OH$ collected during $N_2$ disproportionation reaction, the $NH_2OH$ product of $N_2$ disproportionation reaction was collected into an ITO coated glass bottle filled with the solution of 8-quinolinol probe (Fig. 3a) for 30 min. The concentration of indooxine produced through the reaction between $NH_2OH$ and 8-quinolinol was quantified by UV-Vis experiments (Supplementary Fig. 5a), and the concentration of collected $NH_2OH$ was derived accordingly. Then, under the exactly identical conditions, the $NH_2OH$ product of $N_2$ disproportionation reaction was collected into an ITO coated glass bottle filled with deionized water (without 8-quinolinol). The solution containing $NH_2OH$ was transferred into a cell for electrochemical reduction, which was powered by a 1.5 V solar cell battery (Supplementary Fig. 9). The positive and negative electrodes (diameter 0.5 mm) connected to the 1.5 V battery were immersed into the solution to a depth of 1 cm for 35 min. The formation of $NH_3$ was confirmed by MS analysis (Fig. 4a). The concentration of generated $NH_3$ was spectrophotometrically determined using the indophenol blue method[43]. In detail, a 1 mL of the solution from the electrochemical reduction bottle was transferred into a 15 mL centrifuge tube. Then, 9 mL of 0.005 mol L$^{-1}$ sulfuric acid aqueous solution, 0.5 mL of 50 g L$^{-1}$ salicylic acid solution containing 50 g L$^{-1}$ sodium citrate and 2 mol L$^{-1}$ NaOH solution, 0.1 mL of 0.05 mol L$^{-1}$ NaClO and 0.1 mL of 10 g L$^{-1}$ $Na_2[Fe(NO)(CN)_5]•2H_2O$ were added into the tube and mixed. After 1 h, the absorption spectrum was measured using UV-Vis spectrophotometer (Supplementary Fig. 9). The formation of indophenol blue product was determined by measuring the absorbance at 705 nm. The calibration of the method was achieved using

ammonium chloride solutions of known concentration as standards. The concentration of $NH_3$ was derived from the calibration curve (Supplementary Fig. 9).

## Reaction of HNO with cysteine

The reaction of HNO with cysteine was investigated using the setup shown in Fig. 3a. The experiment was done under the same conditions as in Fig. 3a, with the exception that the P-CM probe solution in the collection bottle was replaced with cysteine solution at different concentrations (1–50 µM). Cystine produced by the reaction between HNO and non-reacted cysteine were detected by MS analysis (Fig. 4c).

## Theoretical calculations

Theoretical calculations were performed using the Gaussian 16 electronic structure programs and ORCA quantum chemistry program package (v5.0.4). The geometries of reactants, intermediates and products were optimized with B2GP-PLYP density functional with quadruple basis set. The same functional was used for vibrational analysis and the Gibbs free energy estimation at 298 K. This virtual orbital-dependent density functional theory method was chosen based on its performance on a set of small, related molecules with available experimental or high-level electronic structure data—namely $N_2$, NO, HNO and $(H_2O)_2^+$. Final electronic energies were calculated at CCSD(T)/CBS level with an expected accuracy of <1 kcal mol[1] (0.04 eV) for single-reference molecular systems. Multi-reference character of the proposed TS state (Fig. 2b), supported by fractional occupation number weighted electron density analysis (Supplementary Fig. 3), precludes us from drawing a precise conclusion about its position on the energy diagram. Consistent with the results of numerous recent benchmark studies, a difference of ≈0.10 eV was observed between CCSD(T)/CBS and B2GP-PLYP/aug-cc-pVQZ data with the exception of $(H_2O)_2^+$ conformations. See Supporting Information for details and references.

## Reporting summary

Further information on research design is available in the Nature Portfolio Reporting Summary linked to this article.

## Data availability

The authors declare that all the data that supports the findings of the study are included in the main text and Supplementary Information Files. The source data are available from the corresponding authors on request. All the data generated in this study have been deposited in the Figshare database under [https://doi.org/10.6084/m9.figshare.25028042]. Source data are provided with this paper.

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

## Acknowledgements

The authors acknowledge Dr. Lili Sun for preliminary calculations. The authors also thank Ms. Xuejiao Chen and Mr. Yuan Zhong for their help in performing the MS experiments. The authors are particularly grateful for the proof reading of the manuscript by Prof. Yinquan Wang. The work was supported by National Natural Science Foundation of China (21520102007, 22104014, 22364002, 22164002) (H.C., Xin.Z., and Xia.Z.), Jiangxi Provincial Natural Science Foundation (20232BAB213047) (Xia.Z.), Jiangxi Province International Cooperation

Project (20203BDH80W010, 20232BBH80012) (H.C.), and Jiangxi University of Chinese Medicine School-level Science and Technology Innovation Team Development Program (CXTD22005, 2004-5252300403). (H.C.).

## Author contributions

H.C. independently developed the idea of the research, supervised the project and prepared the manuscript. Xia.Z. collected the MS data, interpreted the MS data, and co-wrote the manuscript draft. K.C., Xin.Z., W.Z., R.B., and H.C. edited the manuscript. L.H., J.W., W.Z and H.C. designed and conducted the HT22 cells experiments, collected and analyzed the data. Xin.Z., J.L., W.Y., R.S., K.H., S.F., and Xia.Z. collected the UV-Vis, fluorescence spectroscopy, Infrared, Raman and NMR data. R.B. conducted all the calculations.

## Competing interests

The authors declare no competing interests.
