## [Peer Review File · Nature Communications]

Efficient catalyst-free N₂ fixation by water radical cations
under ambient conditionsEditorial Note: Parts of this Peer Review File have been redacted as indicated to remove third-party material where no permission to publish could be obtained.

REVIEWER COMMENTS

Reviewer #1 (Remarks to the Author):

This paper presents an approach and study of molecular nitrogen (N_2) fixation in which a water radical, $(H_2O)_2^+$ formed in a water vapor containing plasma is reacted with N_2 to form two products of interest, HNO and NH_2OH . Fundamental experiments are carried out to characterize these products and intermediates using mass spectrometry. The measurements are supported by modeling of the reaction mechanism. Finally, potential applications of these compounds to make ammonia (NH_3), chemically react and convert cysteine, and serve as a neuroprotective agent are demonstrated.

Overall, the study provides an interesting twist on nitrogen fixation in terms of both the approach and the products. I think such new perspectives are needed in a field that has been focused on NH_3 and a few chemical techniques (thermocatalysis, electrocatalysis, photocatalyst, plasma, etc.) and thus in some ways limited in terms of substantial advancement. For this reason, I think the study should be of interest and make an impact. That being said, there are some issues with the study, including the presentation/organization and the scientific details, that need to be addressed before it can be accepted for publication. I provide detailed comments and questions below:

1. I found the introduction to be misleading. The authors focus on nitrogen fixation to NH_3 which is not the focus of their study. They do show at the end that NH_2OH can be reduced to NH_3 , but the real focus is NH_2OH and HNO. Relatedly, they try to motivate their study by saying that they want to address our dependence on NH_3 . I think this is an incorrect statement. Our dependence on NH_3 is because NH_3 is valuable. It is a very good fertilizer. It is also a great carrier for hydrogen. The authors even say "the prodigious amounts of NH_3 utilized...is dictated by the scarcity of alternative routes of N_2 fixation rather than by the intrinsic usability of NH_3 ." I don't agree with this statement at all! There is no replacement for NH_3 in applications that would be better.

The authors would be much better off writing the introduction about NH_2OH and HNO and their potential uses which they do much later on p. 11 and 12 for example. This verbiage should be moved to the introduction to motivate why making NH_2OH and HNO is interesting.

They should then remove all the text that says we need to stop our dependence on NH_3 which is not the main focus of their study and is not correct.

2. I found the experimental setup, measurements, and conclusions shown in Fig. 1 to be confusing and somewhat inconsistent with others parts of the study. As I understand it, the experiments in Fig. 1 consist of a plasma formed only in Ar and water vapor reacting with a inert N_2 flow (no plasma). Thus, only the water is excited by the plasma and then reacts essentially with ground state N_2 . The authors talk about further activation in the ion trap. Is that part of the mass spectrometer? If so, is there a way to know what happens in the ion trap and what does not (and only happens in the plasma)? Is there a way to remove the ion trap? I found this part very unclear. The authors state on p. 8 "In our experiments

singlet N₂ could be activated to triplet N₂* through collisions with high-energy charged species in the ion source or with accelerated (H₂O)²⁺ species in the ion trap.” How are these differentiated?

Relatedly, did the authors try flowing N₂ in the plasma and doing mass spectrometry? It seems to me 1) that would be valuable to compare with the experiment where N₂ is flowing separately to see how it changes the products and 2) that would allow a better comparison with later scale up experiments shown in Fig. 3 where it looks like N₂ is mixed with H₂O. In fact, after seeing Fig. 1, I was a bit surprised when in Fig. 3, the N₂ is mixed with water because I thought maybe there was something special about keeping N₂ separate from H₂O. After seeing Fig. 3, I am more confused.

3. For the scaled up reaction (as well as mass spectrometry), the authors provide detailed characterization of a few species by spectrophotometric methods such as the NH₂OH and HNO that are of interest. What about other products? What about NH₃/NH₄⁺? NO₃⁻? NO₂⁻? H₂O₂? I would be very surprised if none of these products are formed in a reaction between N₂ and H₂O in a plasma. How are the authors able to produce so much NH₂OH and HNO without further conversion into NH₃ and NO₃⁻ or HNO₃?

Relatedly, the energy consumption or efficiency of the scaled up process and the the solar powered reduction of NH₂OH to NH₃ and how it compares to Haber-Bosch or other reports of electrochemical-based or plasma-based production of NH₃ from N₂ and H₂O should be moved at least as text to the main text (not just in the SI).

4. There are no details provided for the experimental system in Fig. 1. Where are the electrodes for the plasma? What are the materials? Is this DC voltage? Similarly there are no details for the experimental system in Fig. 3a. Is this DC or some other frequency (I assume some other to form 76 plasmas)? What are the electrical characteristics? The analysis should be related to above to show how the energy consumption is calculated. The table in the SI does not provide any details of the power, etc. used to make the calculations.

5. The abstract needs to be revised. It does not include all the results from the study. It does not mention the plasma (water plasma or scaled up reaction with a plasma). It does not mention the applications demonstrated.

6. Finally some grammatical and other minor edits or comments:

a. Please use NO_x everywhere (not NO_x).

b. Abstract: via the distinctive

c. P. 3: for global sustainable development. (remove everything after and...)

d. What does “plasma-based methods currently comes at the cost of N₂ dissociation/burning” mean? How is N₂ burned in a plasma?

e. P. 11: Note that both products

Reviewer #2 (Remarks to the Author):

Activating and converting N₂ directly into value-added molecules might be the most important chemical reaction to human being. In this manuscript, Zhang et al. reported a new strategy of N₂ activation under mild conditions by utilizing water radical cations produced from discharging of water vapor. The water radical cations per se are also very exotic species. They found that N₂ is simultaneously oxidized and reduced into two high-value products, HNO and NH₂OH. This approach offers low cost and high energy efficiency of N₂ fixation: according to the current market, they estimated that 7310 \$ estimate value of HNO and 1.5 \$ estimate value of NH₂OH can be produced at the cost of kWh electricity (~0.2 \$).

Overall, the major breakthrough of this paper is the simultaneous oxidation and reduction of nitrogen without catalyst, which is of interest to the general readership of Nature Communications. I recommend publication of this work.

Reviewer #3 (Remarks to the Author):

In this experimental and computational paper, the authors found that (H₂O)₂⁺ in its two-center-three-electron configuration can specifically activate N₂ via the formation of HONH-HNOH⁺ intermediate to selectively disproportionate it into two high value-added products, HNO and NH₂OH under mild ambient conditions and with no catalyst involved. This result gives a new look and perspectives to the problem of green N₂ fixation. I recommend publication after the authors consider the suggestions below.

1) Fig. 2: In reality, what is of interest will always be the zero point vibrational energy corrected values. So electronic energies, together with Gibbs free energies should be given in Fig. 2. The authors themselves also declared that vibrational frequencies of all the key species were calculated at the same level of theory to obtain their Gibbs free energies at 300 K in the Theoretical calculations section, but no Gibbs free energies are reported in the paper. For the reader's convenience, specific energy values of all the key species should be given in Fig. 2. The authors can refer to some references (J. Am. Chem. Soc. 2020, 142, 13, 6244 and so on).

2) pg 8 line 154: The authors declared that "our calculations indicate that out of the two co-existing (H₂O)₂⁺ structures, i.e., hydrogen-bonded [H₃O⁺...OH] and two-center-three-electron (2c-3e) [H₂O...OH₂]⁺, the latter could effectively react with N₂^{*} (Fig. 2)." But only a possible reaction pathway for triplet N₂^{*} and 2c-3e [H₂O...OH₂]⁺ is reported. Is there a less favorable pathway for triplet N₂^{*} and hydrogen-bonded [H₃O⁺...OH]? This is not mentioned in this paper. On the other hand, both experimental and computational study of ionized water clusters suggested the proton-transferred or called hydrogen-bonded structure is energetically preferred by the (H₂O)₂⁺ clusters (J. Phys. Chem. A 113 (2009) 4772; Phys. Chem. Chem. Phys., 2013, 15, 16214). Thus, the reaction pathway between dominating [H₃O⁺...OH] and triplet N₂^{*} may be welcome.

3) In the Reaction mechanism section, it would be helpful to give the Cartesian coordinates of all optimized structures (or at least intermediates and transition state) in the Supporting Information.

4) The geometries and electronic energies were obtained with B2GP-PLYP density functional. Since energy span can gain marked improvement via a single point energy strategy, it would be helpful to utilize CCSD(T) single point energies in Fig. 2. In alternative to this, the authors can also test relative electronic energy differences between CCSD(T) method and B2GP-PLYP functional.

5) The choice of virtual orbital-dependent functionals was determined by a comparison with available experimental data in literature. The authors tested DFT functionals for singlet and triplet N₂. Do the authors test DFT functionals for other reactant and products (water dimer radical cation, HNO, and NH₂OH) if possible?

6) “The INT1* structure is then converted into the excited-state intermediate HONH-HNOH+•* by the direct double-proton transfer through TS* structure (Fig. 2).” Is the new mechanism of N₂ fixation confirmed by intrinsic reaction coordinate (IRC) analysis? If yes, which method is used?

7) “Transition state energy was estimated from the combination of B2PLYP and M06-2X data.” This sentence is a little confusing.

8) Give references for the basis set used.

Reviewer #4 (Remarks to the Author):

The manuscript “Efficient catalyst-free N₂ fixation by water under ambient conditions” by Zhang and coworkers describes a reaction that can produce HNO and NH₂OH from N₂ via reaction with the water dimer radical cation. The authors claim that this reaction is a good way to “fix” N₂ and therefore may have important utility. However, there are some concerns with some of the claims in this article that need to be reconciled. These are listed below:

1. The generation of HNO and NH₂OH from N₂ disproportionation via this chemistry is intriguing.

However, the authors fail to mention that HNO (a potent electrophile) reacts rapidly with NH₂OH (a potent nucleophile) to give N₂ and 2H₂O. This rapid and favorable reaction could negate the utility of this chemistry as sources for either HNO or NH₂OH.

2. Also, HNO is a very fleeting species due to self-reactivity to give N₂O and water. Therefore, in experiments where this chemistry was used as a source of HNO (e.g., the cell work), it needs to be realized that HNO is present for only a very short period (if at all) – thus, prolonged incubation is really meaningless.

3. Moreover, considering the propensity for HNO to react with and inhibit thiol proteins, it is difficult to

understand how pre-incubation with HNO will protect cells from H₂O₂ toxicity since much of H₂O₂ toxicity is thought to be combatted using thiol proteins. Indeed, H₂O₂ toxicity is also thought to involve modification of thiol proteins.

4. Also, it is interesting that HNO treatment of cells results in increased "viability". The assay used actually measures the reducing capacity of the cell (not specifically viability). Although cell viability and reducing capability may correlate, this is not strictly viability. Regardless, some comment on what HNO may be doing is warranted here, especially since this is an unexpected finding.

Reviewer #5 (Remarks to the Author):

In their manuscript entitled "Efficient catalyst-free N₂ fixation by water under ambient conditions", Xiaoping Zhang et al. report the possibility to form nitroxyl and hydroxylamine by the reaction of N₂ with water dimer radical cations generated by plasma. The products are first observed by mass spectrometry and the process has then been transferred to a larger array-like reactor for wet chemical analysis. A possible reaction mechanism that sounds overall exotic but reasonable to me is derived with support from theoretical calculations. First of all, I can say that I read this manuscript with high interest and it is well written. Plasma-assisted N₂ fixation has been studied for a long time but I think that what is shown in the present submission is indeed a novel reaction pathway in this field (without plasma from N₂) which could be energetically more favourable than the state-of-the art methods as it is based on water vapor plasma. On the other side, the results shown here are that unexpected and unconventional that more control experiments are needed to fully support the conclusions. For the large related community working on electrocatalytic and photocatalytic N₂ fixation a few benchmark numbers are also missing that should be added. I recommend that the following experiments should be done and points should be clarified before possible publication in Nature Communications:

1) In the plasma community the "conversion rate" of N₂ (in %) is an often used benchmark number. How is this conversion rate in this case?

2) It is good that the reaction was performed in a scaled up system where standard wet-chemical analysis becomes possible. I would expect authors to also perform these experiments with ¹⁵N and do NMR analysis of the formed products with a quantitative comparison of products formed from ¹⁵N and ¹⁴N.

3) N₂ with 99.99% purity is not very pure. It would be an expected reference experiment to simply use a 99.999% nitrogen source for comparison. Another very much recommended reference experiment would be the use of sythetic air.

4) In my opinion the chapter about the "Application of the reaction products" is not needed. The first part about ammonia formation might be but the rest should be removed which would authors give

space for more experimental validation of their proposed mechanism.

5) What would a FE of 20% mean in this case? Is there any side product of the reaction? I am also missing a comparative discussion of the overall energy input that would be needed to make, say, a mol of ammonia and the comparison to other plasma-based methods as well as H-B and electrochemical/photochemical N₂ fixation.

6) What would be the estimated ammonia production rate?

Other points:

- The title of the work will attract attention but should somewhere contain the word plasma. It is otherwise incomplete and misleading.

Please note that all page numbers, line numbers, reference numbers, etc. in this list refer to the *revised manuscript*.

REVIEWER COMMENTS

Reviewer #1 (Remarks to the Author):

Comments to the Author

This paper presents an approach and study of molecular nitrogen (N_2) fixation in which a water radical, $(H_2O)_2^+$ formed in a water vapor containing plasma is reacted with N_2 to form two products of interest, HNO and NH_2OH . Fundamental experiments are carried out to characterize these products and intermediates using mass spectrometry. The measurements are supported by modeling of the reaction mechanism. Finally, potential applications of these compounds to make ammonia (NH_3), chemically react and convert cysteine, and serve as a neuroprotective agent are demonstrated.

Overall, the study provides an interesting twist on nitrogen fixation in terms of both the approach and the products. I think such new perspectives are needed in a field that has been focused on NH_3 and a few chemical techniques (thermocatalysis, electrocatalysis, photocatalyst, plasma, etc.) and thus in some ways limited in terms of substantial advancement. For this reason, I think the study should be of interest and make an impact. That being said, there are some issues with the study, including the presentation/organization and the scientific details, that need to be addressed before it can be accepted for publication. I provide detailed comments and questions below:

⇒ We appreciate the reviewer's positive and insightful suggestions. We have modified the presentation/organization (revised Introduction, Abstract, added comparison on energy consumption or efficiency, etc.) and the scientific details (revised Fig. 1 and Fig. 3, added details on experiments, characterization of side products) of this manuscript as suggested. We hope a clear and well-organized revised manuscript fully address all the concerns raised by the reviewer.

1. I found the introduction to be misleading. The authors focus on nitrogen fixation to NH_3 which is not the focus of their study. They do show at the end that NH_2OH can be reduced to NH_3 , but the real focus is NH_2OH and HNO . Relatedly, they try to motivate their study by saying that they want to address our dependence on NH_3 . I think this is an incorrect statement. Our dependence on NH_3 is because NH_3 is valuable. It is a very good fertilizer. It is also a great carrier for hydrogen. The authors even say “the prodigious amounts of NH_3 utilized...is dictated by the scarcity of alternative routes of N_2 fixation rather than by the intrinsic usability of NH_3 .” I don’t agree with this statement at all! There is no replacement for NH_3 in applications that would be better.

The authors would be much better off writing the introduction about NH_2OH and HNO and their potential uses which they do much later on p. 11 and 12 for example. This verbiage should be moved to the introduction to motivate why making NH_2OH and HNO is interesting. They should then remove all the text that says we need to stop our dependence on NH_3 which is not the main focus of their study and is not correct.

⇒ **We thank the reviewer for raising this point. We have revised the Introduction accordingly. The focus has been shifted toward NH_2OH and HNO and their potential uses. The text that said we need to stop our dependence on NH_3 has been removed. The revised Introduction (see Pages 3-4, Lines 37-74) reads as follows:** “Nitrogen is an essential element for all living organisms on our planet. Molecular nitrogen (N_2) accounts for more than 99% of global nitrogen ¹ but is extremely chemically stable ($\text{N}\equiv\text{N}$ bond energy ca. 9.5 eV) and thus cannot be directly utilized unless fixed by alternating its oxidation state into bioavailable forms ². The problem of N_2 fixation is one of the most important for sustainable chemistry. Currently, N_2 on Earth is predominantly fixed through geochemical processes such as lightning, biologically by nitrogenases, and industrially through the Haber-Bosch (HB) process ³. While HB is currently the major industrial process for N_2 fixation to ammonia (NH_3), it is associated with intensive reaction conditions (ca. 100 bar and 500°C), severe environmental pollution (>1% of the global carbon emission) and high consumption of fossil fuel (1%–2% of global

energy consumption). These issues are becoming increasingly crucial for the global sustainable development, and urge novel strategies for N₂ fixation under mild conditions ^{4,5}. Extensive research is being done in search of alternative strategies for N₂ fixation, including electrocatalytic ^{6,7}, photocatalytic ⁸, biological ⁹, and plasma-based ¹⁰ methods, but none of these methods have yet been able to rival the overall performance of HB process with regard to the cost efficiency, scalability and selectivity of N₂ fixation ^{6,11,12}.

Recent studies indicate that the N≡N bond can be weakened by accepting electrons from the bonding orbitals of N₂ to the antibonding orbitals and/or donating electrons, which would make its functionalization feasible ^{13,14}. The weakening of N≡N bond could be further promoted through the excitation of N₂ into its triplet state, *e.g.*, by electronic or collisional activation with molecules or ions ^{15,16}. Recently, we have discovered that abundant radical cations of water clusters, especially in the dimer form, (H₂O)₂⁺, can be produced by depositing suitable amounts of energy into the pure water vapor at atmospheric pressure ¹⁷. The as-prepared water radical cations showed the high reactivity toward a wide range of volatile molecules, such as benzene, acetone, ethyl acetate and dimethyl disulphide, revealing rich chemistry with the ionic and radical characters ¹⁷⁻²⁰.

In this work, we discovered that, owing to its distinct two-center-three-electron (2c-3e) configuration, (H₂O)₂⁺ can specifically activate N₂ via the formation of HONH-HNOH⁺ intermediate to selectively disproportionate it into hydroxylamine (NH₂OH) and nitroxyl (HNO) products under mild ambient conditions and with no catalyst involved. These products are not commonly observed upon N₂ fixation and have high value. NH₂OH is widely used in medicine, textile industry, electronics, chemical synthesis, nuclear industry and other fields ²¹. HNO is a valuable material for medical and biology utilities, particularly for biological targets of thiols and metalloproteins in fighting cancer ²². Importantly, HNO is known for its cardioprotective and neuroprotective effects and is resistant to superoxide

scavenging and tolerance development²³. Overall, the study provides an interesting twist on N₂ fixation in terms of both the approach and the products.”.

2. I found the experimental setup, measurements, and conclusions shown in Fig. 1 to be confusing and somewhat inconsistent with others parts of the study. As I understand it, the experiments in Fig. 1 consist of a plasma formed only in Ar and water vapor reacting with a inert N₂ flow (no plasma). Thus, only the water is excited by the plasma and then reacts essentially with ground state N₂. The authors talk about further activation in the ion trap. Is that part of the mass spectrometer? If so, is there a way to know what happens in the ion trap and what does not (and only happens in the plasma)? Is there a way to remove the ion trap? I found this part very unclear. The authors state on p. 8 “In our experiments singlet N₂ could be activated to triplet N₂^{*} through collisions with high-energy charged species in the ion source or with accelerated (H₂O)₂⁺ species in the ion trap.” How are these differentiated?

⇒ **We thank the reviewer for raising this point. We apologize for the confusing description of experimental process in the original Fig. 1. In the revised manuscript we have modified Fig. 1 as shown below. The experimental results in Fig. 1a-b and Fig. 1c-d correspond to the two different series of experiments.** Fig. 1a displays the disproportionation reaction of N₂ by water plasma at atmospheric pressure with online mass spectrometry detection. In Fig. 1c, (H₂O)₂⁺ formed by water plasma is selectively isolated inside the ion trap and is then specifically reacted with N₂ gas leaked into the ion trap. The formed products are then directly detected inside the same ion trap. The role of experiments in Fig. 1c is to confirm the mechanism of N₂ disproportionation in water plasma by the reaction with (H₂O)₂⁺ species though the HONH-HNOH⁺ intermediate.

[REDACTED]

Fig. 1 | Disproportionation reaction of N₂ with water dimer radical cation. a Experimental setup to study the interaction of N₂ with water vapor plasma at ambient conditions. Stainless-steel discharge needle was used as electrode. DC: direct current. **b** The corresponding mass spectrum of ionic products in Fig. 1a. Red-color marks correspond to the products specific to the reaction between water vapor plasma and N₂. **c** Ion trap reactor applied to study the reaction between N₂ and isolated (H₂O)₂⁺ (*m/z* 36) in vacuum. **d** The corresponding mass spectrum of ionic products in **c**. **e** Ionic fragments of the reaction intermediate at *m/z* 64 induced by collisional activation inside the ion trap. **f** Mass spectrum of the ionic species observed during the interaction between water vapor plasma and ¹⁵N₂ (¹⁵N₂ gas instead of ¹⁴N₂ in **a**). Red-color marks correspond to the products specific to the reaction between water vapor plasma and ¹⁵N₂.

⇒ The “activation in the ion trap” was used to refer to the acceleration of (H₂O)₂⁺ ions by radio frequency field in the ion trap. The kinetic energy of the

accelerated $(\text{H}_2\text{O})_2^{+\bullet}$ ions is transferred to the electronic excitation of N_2 through collisions between $(\text{H}_2\text{O})_2^{+\bullet}$ and N_2 in the ion trap. We have revised the experimental description and the content of Fig.1 for better clarity as follows: “In our first experiment, water plasma was generated by discharge of water/argon vapor mixture (Supplementary Fig. 1a). The major ionic species observed by real-time mass spectrometry detection included protonated water clusters, $(\text{H}_2\text{O})_2\text{H}^+$ (m/z 37) and $(\text{H}_2\text{O})_3\text{H}^+$ (m/z 55), as well as abundant water dimer radical cation, $(\text{H}_2\text{O})_2^{+\bullet}$ (m/z 36) (Supplementary Fig. 1b), in agreement with previous reports^{17-20,24,25}. Remarkably, when N_2 was flown into the discharge area (carrier argon gas was replaced by N_2 : Fig. 1a), abundant ions at m/z 32 and m/z 33 and m/z 64 were additionally observed (Fig. 1b). Note that the same signals were observed at similar intensity when N_2 was introduced to intersect with the water plasma ca. 1 cm away from the discharge area (Supplementary Fig. 1c, 1d). These observations indicate that the signals m/z 32, m/z 33 and m/z 64 correspond to the species formed due to the interaction between water plasma and neutral N_2 . We tentatively assigned these signals to HNOH^+ (m/z 32), $\text{NH}_2\text{OH}^{+\bullet}$ (m/z 33) and $\text{HONH-HNOH}^{+\bullet}$ (m/z 64), respectively. We speculated that these new ionic species could be derived through the reaction between $(\text{H}_2\text{O})_2^{+\bullet}$ and N_2 . To verify this assumption, $(\text{H}_2\text{O})_2^{+\bullet}$ ions (m/z 36) were selectively isolated in the ion trap by radio frequency (RF) field with peak-to-peak voltage of ~ 100 V in the presence of neutral N_2 gas, which was directly introduced into the ion trap (Fig. 1c). When the trapped $(\text{H}_2\text{O})_2^{+\bullet}$ ions were activated by RF field, the product signals were unambiguously observed at m/z 18 (elimination of H_2O from $(\text{H}_2\text{O})_2^{+\bullet}$), m/z 19 (elimination of $\bullet\text{OH}$ from $(\text{H}_2\text{O})_2^{+\bullet}$), m/z 33 ($\text{NH}_2\text{OH}^{+\bullet}$), m/z 51 ($\text{NH}_2\text{OH}^{+\bullet}$ plus H_2O), m/z 55 ($(\text{H}_2\text{O})_3\text{H}^+$) and m/z 64 ($\text{HONH-HNOH}^{+\bullet}$) (Fig. 1d). These observations confirm the occurrence of reaction between $(\text{H}_2\text{O})_2^{+\bullet}$ and N_2 . Curiously, the signal of HNOH^+ abundantly produced upon the interaction between water plasma and N_2 (m/z 32 in Fig. 1b) was almost undetectable when $(\text{H}_2\text{O})_2^{+\bullet}$ was activated in the ion trap (Fig. 1d), probably because the HNO was initially

created as neutral species. During the interaction between water plasma and N_2 , neutral HNO could be easily protonated by other ionic species such as $(H_2O)_2H^+$ (m/z 37) to give the protonated signal at m/z 32 (Fig. 1b), whereas when produced in the ion trap neutral HNO would be instantly pumped out of the ion trap (maintained at high vacuum of 10^{-5} Torr).” (see Pages 5-7, Lines 89-117)

⇒ The experiments shown in Fig. 1a and Fig. 1c correspond to the two different series of experiments. The experiment in Fig. 1a is in the ion source (atmospheric pressure) part of the mass spectrometer, while the experiment in Fig. 1c is in the ion trap (vacuum) part of the mass spectrometer. Thus, the content of “In our experiments singlet N_2 could be activated to triplet N_2^* through collisions with high-energy charged species in the ion source or with accelerated $(H_2O)_2^+$ species in the ion trap.” has been replaced as “When the disproportionation of N_2 is carried in water plasma (Fig. 1a) singlet N_2 could be easily activated to triplet N_2^* through collisions with high-energy $(H_2O)_2^{+*}$ and other species in water plasma. When the reaction between N_2 and $(H_2O)_2^{+*}$ is carried inside the ion trap, singlet N_2 could be activated to triplet N_2^* through collisions with $(H_2O)_2^{+*}$ species activated by RF field (Fig. 1c). Accordingly, no reaction products were observed when $(H_2O)_2^{+*}$ ions were trapped in N_2 gas without activation (Supplementary Fig. 1i). Also, note that, being an electronic transition, N_2 activation to N_2^* occurs on a much shorter time scale compared to atomic rearrangements. Therefore, the event of N_2 activation to N_2^* and the following association between N_2^* and $(H_2O)_2^{+*}$ could occur within a single collision between N_2 and $(H_2O)_2^{+*}$.” (see Page 9, Lines 161-170)

[REDACTED]

Fig. S1. Reference experiments with regard to the reaction between N₂ and water radical cations shown in Fig. 1a. **a** Schematic illustration of the experiment in which N₂ in Fig. 1a was replaced by Ar. **b** Mass spectrum recorded in the experiments shown in Supplementary Fig. 1a was replaced by Ar. **c** Experimental two-channel setup to study the interaction of neutral N₂ with water radical cations by online mass spectrometry. **d** Mass spectrum recorded in Supplementary Fig. 1c. **e** Dissociation pathways of the HONH-HNOH⁺⁺ intermediate m/z 64 formed under the conditions of Fig. 1a. **f** Mass spectrum recorded when H₂O in the experiments shown in Fig. 1a was replaced by D₂O, other experimental conditions being the same. **g** Tandem mass spectra of the HONH-HNOH⁺⁺-type intermediates revealed in the disproportionation reaction of N₂ with (H₂O)₂⁺⁺ using different isotopic substitutes: m/z 68 (DOND-DNOD⁺⁺) and m/z 66 (HO¹⁵NH-H¹⁵NOH⁺⁺). **h** Mass spectrum of N₂⁺⁺ (corresponding signal at m/z 28) in the presence of neutral water vapor in the ion trap. No signals at m/z 64 (HONH-HNOH⁺⁺), m/z 32 (HNOH⁺) or m/z 33 (NH₂OH⁺⁺) could be observed under any experimental conditions tested, even by applying collision energy to the isolated N₂⁺⁺ ions (isolation width of 5 Da; the activation Q value of 0.4). **i** Isolation of ions at m/z 36 inside the ion trap with zero collisional activation energy.

Relatedly, did the authors try flowing N₂ in the plasma and doing mass spectrometry? It seems to me 1) that would be valuable to compare with the experiment where N₂ is flowing separately to see how it changes the products and 2) that would allow a better comparison with later scale up experiments shown in Fig. 3 where it looks like N₂ is mixed with H₂O. In fact, after seeing Fig. 1, I was a bit surprised when in Fig. 3, the N₂ is mixed with water because I thought maybe

there was something special about keeping N₂ separate from H₂O. After seeing Fig. 3, I am more confused.

⇒ We thank the reviewer for raising this point. In fact, we have tested these two modes of analysis: when N₂ is flown separately (Supplementary Fig. 1c, 1d above) and when N₂ is flown directly into the water plasma (Fig. 1a above). No significant difference was observed. Originally, we chose to display the setup where N₂ and H₂O are flown separately in order to emphasize that the reaction occurs between neutral N₂ and water plasma. However, we agree with the reviewer that this brings inconsistency regarding the follow-up description of scaled-up experiment. Therefore, we have revised the manuscript, including the content of Fig. 1a, as follows: “Remarkably, when N₂ was flown into the discharge area (carrier argon gas was replaced by N₂: Fig. 1a), abundant ions at m/z 32 and m/z 33 and m/z 64 were additionally observed (Fig. 1b). Note that the same signals were observed at similar intensity when N₂ was introduced to intersect with the water plasma ca. 1 cm away from the discharge area (Supplementary Fig. 1c, 1d). These observations indicate that the signals m/z 32, m/z 33 and m/z 64 correspond to the species formed due to the interaction between water plasma and neutral N₂. We tentatively assigned these signals to HNOH⁺ (m/z 32), NH₂OH⁺ (m/z 33) and HONH-HNOH⁺ (m/z 64), respectively. We speculated that these new ionic species could be derived through the reaction between (H₂O)₂⁺ and N₂.” (see Page 6, Lines 93-102)

3. For the scaled up reaction (as well as mass spectrometry), the authors provide detailed characterization of a few species by spectrophotometric methods such as the NH₂OH and HNO that are of interest. What about other products? What about NH₃/NH₄⁺? NO₃⁻? NO₂⁻? H₂O₂? I would be very surprised if none of these products are formed in a reaction between N₂ and H₂O in a plasma. How are the authors able to produce so much NH₂OH and HNO without further conversion into NH₃ and NO₃⁻ or HNO₃?

⇒ **We thank the reviewer for raising this point. As per the reviewer’s comment, we have also characterized other products in the reaction, including NH_4^+ , NO_3^- , NO_2^- and H_2O_2 (see Figs. below). The following text has been added in the revised manuscript “Thus, in supplementary mass spectrometry (Supplementary Fig. 13a-d), ion chromatography (Supplementary Fig. 13e-f), and UV-vis absorption spectroscopy (Supplementary Fig. 14) we also observed NO_3^- , NO_2^- and H_2O_2 , products, which were likely produced by further conversion of the HNO product. The content of NO_3^- , NO_2^- and H_2O_2 produced over 10 min was estimated to be about 0.007 mM, 0.004 mM, and 0.03 mM, respectively. Interestingly, ion chromatography (Supplementary Fig. 15) and ^1H NMR spectroscopy (Supplementary Fig. 7b) data also indicated the production of NH_4^+ (about 0.02 mM over 10 min), which is likely through the reduction of NH_2OH product. The simple reduction of NH_2OH to NH_3 is demonstrated by the results in Fig. 4a. The origin of NH_3 via the reduction of N_2 by water plasma was further confirmed by isotope-labeling experiments (Supplementary Fig. 7b).”** (see Pages 15-16, Lines 302-312)

⇒ **The following description has been added in the method section:**
“Quantification of NH_3 , NO_3^- and NO_2^- with ion chromatography. The amounts of NH_3 , NO_3^- and NO_2^- formed over the interaction between water plasma and N_2 were also estimated by ion chromatography (Thermo-Fisher DIONEX ICS-1100). The test procedure followed the standard operation procedure provided by the vendor. Briefly, 25 μL of the resultant was injected into the chamber for separation. Delivery speed was 4 mL min^{-1} . The peak eluted at the retention time of 9 min was assigned to NH_4^+ . The amount of NH_4^+ was estimated using a calibration curve by standard solutions (Supplementary Fig. 15). The peaks eluted at the retention time of 5 min and 8 min were assigned to NO_2^- and NO_3^- . The amounts of NO_2^- and NO_3^- were estimated using a series of standard solutions (Supplementary Fig. 13e-f).” **and** “Quantification of H_2O_2 by UV-Vis spectroscopy. The amount of H_2O_2 was measured by UV-Vis absorption spectroscopy (UV-1900i, Shimadzu, Japan).

According to reference ⁶⁵, potassium permanganate was reduced by hydrogen peroxide in sulfuric acid with the maximum absorption peak at 525 nm. The amount of H₂O₂ was derived from the amount of consumed MnO₄⁻ for calibration with a series of standard solutions (Supplementary Fig. 14).” (see Pages 24-25, Lines 481-495)

Fig. S13. Characterization of NO₂⁻ and NO₃⁻ products by mass spectrometry (a-d) and ionic chromatography (e-h): **a** online detection of produced species in negative ion detection mode. **b** reference spectrum of standard compounds. **c** blank reference spectrum; **d** calibration curve for the quantitative evaluation of NO₂⁻ and NO₃⁻ products. **e** ion chromatogram of the collected reaction gas. **f** ionic chromatogram of standard compounds. **g** reference blank ion chromatogram. **h** calibration curve.

Fig. S14. Characterization of H₂O₂ product by UV-Vis spectroscopy.

Fig. S15. Characterization of NH₄⁺ by ionic chromatography.

Relatedly, the energy consumption or efficiency of the scaled up process and the solar powered reduction of NH₂OH to NH₃ and how it compares to Haber-Bosch or other reports of electrochemical-based or plasma-based production of NH₃ from N₂ and H₂O should be moved at least as text to the main text (not just in the SI).

⇒ We thank the reviewer for raising this point. The corresponding content of Table S1 and Table S2 has been moved to the main body of the revised manuscript. The following text has been added in the revised manuscript: “In

comparison with the HB process and other methods of N₂ fixation, including catalytic and plasma methods (Tables 1 and 2), our method offers considerably higher economy efficiency (1.54 \$ kWh⁻¹ for NH₂OH and 7310 \$ kWh⁻¹ for HNO), which is owing to the high value of NH₂OH and particularly HNO products as compared to NH₃. In terms of energy cost (g kWh⁻¹), our method is currently ca. 3 orders of magnitude less efficient than HB and ca. 1-2 orders of magnitude less efficient than plasma-based methods for N₂ fixation reported earlier (0.154 g kWh⁻¹ for NH₂OH and 0.149 g kWh⁻¹ for HNO). The higher energy cost in our method is mainly due to the avoidance of a catalyst, high pressures and high temperatures. The energy cost is expected to reduce as the plant continues to be upgraded. In comparison with several catalytic methods for N₂ fixation under mild conditions (near room temperature and atmospheric pressure) reported earlier, our method offers similar yield of N-containing products (1.14 μg·cm⁻²·h⁻¹ for NH₂OH and 0.37 μg·cm⁻²·h⁻¹ for HNO).” (see Pages 16-17, Lines 320-331)

.4. There are no details provided for the experimental system in Fig. 1. Where are the electrodes for the plasma? What are the materials? Is this DC voltage?

Similarly there are no details for the experimental system in Fig. 3a. Is this DC or some other frequency (I assume some other to form 76 plasmas)? What are the electrical characteristics? The analysis should be related to above to show how the energy consumption is calculated. The table in the SI does not provide any details of the power, etc. used to make the calculations.

⇒ We thank the reviewer for pointing out these problems. In Fig. 1a, the electrode for plasma is a discharge needle. The discharge needle is made of stainless steel. The direct current (DC) positive voltage in Fig. 1a was applied. For the revised Fig. 1a, please refer to the Response 2 above. The content of “To build the ion source, a sharp stainless-steel needle (OD 150 μm; end curvature radius ~30 μm) was used as the discharge electrode for plasma. The needle was coaxially inserted into a fused silica capillary (ID 0.25 mm, OD 0.40 mm) and was fixed coaxially with a union tee and silica ferrule. The back end of the tee was

connected to the H₂O/Ar line. The distance from the inlet of the mass spectrometer capillary to the tip of the ion source was 20 mm. The direct current (DC) voltage of +2.0 kV was applied to the stainless-steel needle in order to generate ambient corona discharge.” has been added in the revised manuscript. (see Pages 20-21, Lines 398-405)

⇒ In Fig. 3a (see below), positive DC voltage is also applied in the experimental system. The content of “The 76 anodes of the array were connected to the same positive terminal of the DC high voltage power.” has been added in the revised manuscript. (see Page 21, Lines 414-416)

⇒ The detailed calculations regarding the power, energy cost, etc. have been added in the revised Supplementary information as follows: The energy cost is calculated according to the equations: the power $W = U \times I \times t$ (1) and the energy cost $= W/n$ (2), where W is the power, U is the discharge voltage, I is the ion current, t is the reaction time, n is the molar mass of the product. For example, at $U = 4.2$ kV, $I = 0.031$ mA, $t = 10$ min the amount of produced NH₂OH is ca. 3.372 μg (about 0.102 μmol). The power consumed over 10 min is about 4.2 kV × 0.031 mA × 1/6 h = 78.12 J. The energy cost is about 78.12 J/0.102 μmol = 770 MJ/mol. (see Page 3, Lines 59-64, in supplementary information)

[REDACTED]

Fig. 3a | Schematic illustration of the reaction assembly for scale-up reaction and the collection of reaction products. The 76 anodes of the array were connected to the same positive terminal of the DC high voltage power.

5. The abstract needs to be revised. It does not include all the results from the study. It does not mention the plasma (water plasma or scaled up reaction with a plasma). It does not mention the applications demonstrated.

⇒ **We thank the reviewer for pointing out these problems. The abstract** “The growth and sustainable development of humanity is heavily dependent upon the process of fixing nitrogen (N_2) to ammonia (NH_3). However, the currently adopted methods are associated with severe environmental hazards and tremendous energy costs, which limit their sustainability and profitability. Herein we discovered a catalyst-free disproportionation reaction of N_2 by water dimer radical cation, $(H_2O)_2^{+*}$, which occurs under mild ambient conditions *via* distinctive HONH-HNOH $^{+*}$ intermediate to yield economically valuable nitroxyl (HNO) and hydroxylamine (NH_2OH) products, in alternative to NH_3 . Calculations suggest that the reaction is prompted by the coordination of electronically excited N_2 with $(H_2O)_2^{+*}$ in its two-center-three-electron (2c-3e) configuration. Subsequent excited-state double proton transfer leads to one-step water addition to N_2 . The ambient fixation of N_2 into HNO and NH_2OH with high selectivity offers great profitability and total avoidance of polluting emissions, such as CO_2 or NO_y , thus giving an entirely new look and perspectives to the problem of green N_2 fixation.” **has been changed into** “The growth and sustainable development of humanity is heavily dependent upon N_2 fixation. Herein we discovered ambient catalyst-free disproportionation of N_2 by water plasma which occurs *via* the distinctive HONH-HNOH $^{+*}$ intermediate to yield economically valuable nitroxyl (HNO) and hydroxylamine (NH_2OH) products. Calculations suggest that the reaction is prompted by the coordination of electronically excited N_2 with water dimer radical cation, $(H_2O)_2^{+*}$, in its two-center-three-electron configuration. The reaction products were collected in a 76-needle array discharge reactor with product yields of $1.14 \mu g \text{ cm}^{-2} \text{ h}^{-1}$ for NH_2OH and $0.37 \mu g \text{ cm}^{-2} \text{ h}^{-1}$ for HNO. Potential applications of these compounds were demonstrated to make ammonia (for NH_2OH), as well as to chemically react and convert cysteine, and serve as a

neuroprotective agent (for HNO). The conversion of N₂ into HNO and NH₂OH by water plasma could offer great profitability and reduction of polluting emissions, thus giving an entirely new look and perspectives to the problem of green N₂ fixation.” in the revised manuscript. (see Page 2, Lines 24-35)

6. Finally some grammatical and other minor edits or comments:

- a. Please use NO_x everywhere (not NO_X).
- b. Abstract: via the distinctive
- c. P. 3: for global sustainable development. (remove everything after and...)
- d. What does “plasma-based methods currently comes at the cost of N₂ dissociation/burning” mean? How is N₂ burned in a plasma?
- e. P. 11: Note that both products

⇒ We thank the reviewer for pointing out these questions. a) The content of “NO_y” has been changed into “NO_x” in the revised manuscript.

b) The content of “via distinctive” has been changed into “via the distinctive” in the revised manuscript.

c) The content of “and urge novel strategies for N₂ fixation under mild conditions” has been removed in the revised manuscript.

d) We have changed the inappropriate description. The sentence of “plasma-based methods currently comes at the cost of N₂ dissociation/burning” has been removed in the revised manuscript.

e) The content of “Note that the both products” has been changed into “Note that both NH₂OH and HNO products” in the revised manuscript.

Reviewer #2 (Remarks to the Author):

Activating and converting N₂ directly into value-added molecules might be the most important chemical reaction to human being. In this manuscript, Zhang et al. reported

a new strategy of N_2 activation under mild conditions by utilizing water radical cations produced from discharging of water vapor. The water radical cations per se are also very exotic species. They found that N_2 is simultaneously oxidized and reduced into two high-value products, HNO and NH_2OH . This approach offers low cost and high energy efficiency of N_2 fixation: according to the current market, they estimated that 7310 \$ estimate value of HNO and 1.5 \$ estimate value of NH_2OH can be produced at the cost of kWh electricity (~0.2 \$).

Overall, the major breakthrough of this paper is the simultaneous oxidation and reduction of nitrogen without catalyst, which is of interest to the general readership of Nature Communications. I recommend publication of this work.

⇒ **We thank the reviewer for the positive opinion.**

Reviewer #3 (Remarks to the Author):

In this experimental and computational paper, the authors found that $(H_2O)_2^{+}$ in its two-center-three-electron configuration can specifically activate N_2 via the formation of $HONH-HNOH^{+}$ intermediate to selectively disproportionate it into two high value-added products, HNO and NH_2OH under mild ambient conditions and with no catalyst involved. This result gives a new look and perspectives to the problem of green N_2 fixation. I recommend publication after the authors consider the suggestions below.

⇒ **We thank the reviewer for his/her positive and insightful suggestions.**

1) Fig. 2: In reality, what is of interest will always be the zero point vibrational energy corrected values. So electronic energies, together with Gibbs free energies should be given in Fig. 2. The authors themselves also declared that vibrational frequencies of all the key species were calculated at the same level of theory to obtain their Gibbs free energies at 300 K in the Theoretical calculations section, but no Gibbs free energies are reported in the paper. For the reader's convenience, specific energy values of all the key species should be given in Fig. 2. The authors can refer to some

references (J. Am. Chem. Soc. 2020, 142, 13, 6244 and so on).

⇒ We thank the reviewer for the comments. Fig. 2 (see below) was updated to include Gibbs free energies. We particularly thank the reviewer for providing references to the insightful papers on nitrogen reduction via tetraboration and Borden's study of thermodynamics of N₂ addition reactions, which we have overlooked during our preliminary literature analysis. This reference has been cited in the revised manuscript (see Ref 14).

Fig. 2 | Mechanism and calculation results for the reaction of N₂ with (H₂O)₂⁺. **a** Schematic diagram summarizing a possible mechanism for the reaction of N₂ with (H₂O)₂⁺. **b** The geometries and energies (in eV at 298 K and 1 atm pressure) of possible molecular and ionic species involved in the disproportionation reaction N₂ + (H₂O)₂⁺ → NH₂OH⁺ + HNO calculated with CCSD(T) method. Our expected accuracy is 0.04 eV, with the exception of TS structure (grey, see Supplementary information). Vertical arrows correspond to the process of electronic excitation/de-excitation. a: [H₂O•••OH₂]⁺. p1: NH₂OH⁺. p2: HNO.

2) pg 8 line 154: The authors declared that “our calculations indicate that out of the two co-existing (H₂O)₂⁺ structures, i.e., hydrogen-bonded [H₃O⁺•••OH] and two-center-three-electron (2c-3e) [H₂O•••OH₂]⁺, the latter could effectively react with N₂^{*} (Fig. 2).” But only a possible reaction pathway for triplet N₂^{*} and 2c-3e [H₂O•••OH₂]⁺ is reported. Is there a less favorable pathway for triplet N₂^{*} and hydrogen-bonded [H₃O⁺•••OH]? This is not mentioned in this paper. On the other hand, both experimental and computational study of ionized water clusters suggested the proton-transferred or called hydrogen-bonded structure is energetically preferred by the (H₂O)₂⁺ clusters (J.

Phys. Chem. A 113 (2009) 4772; Phys. Chem. Chem. Phys., 2013, 15, 16214). Thus, the reaction pathway between dominating $[\text{H}_3\text{O}^+\cdots\text{OH}]$ and triplet N_2^* may be welcome.

⇒ **We thank the reviewer for pointing out this problem. The text “our calculations indicate that out of the two co-existing $(\text{H}_2\text{O})_2^{+*}$ structures, i.e., hydrogen-bonded $[\text{H}_3\text{O}^+\cdots\text{OH}]$ and two-center-three-electron (2c-3e) $[\text{H}_2\text{O}\cdots\text{OH}_2]^+$, the latter could effectively react with N_2^* (Fig. 2).” was modified as “In agreement with previous theoretical and experimental reports, our calculations indicate the co-existence of two $(\text{H}_2\text{O})_2^{+*}$ configurations, i.e., hydrogen-bonded $[\text{H}_3\text{O}^+\cdots\text{OH}]$ and two-center-three-electron (2c-3e) $[\text{H}_2\text{O}\cdots\text{OH}_2]^+$ (+0.3 eV)¹⁷. Despite the $[\text{H}_3\text{O}^+\cdots\text{OH}]$ configuration being the global energy minimum for $(\text{H}_2\text{O})_2^{+*}$, we could not find a stable intermediate structure for the binding of N_2^* with $[\text{H}_3\text{O}^+\cdots\text{OH}]$. In contrast, we could easily locate a stable intermediate structure for the binding of N_2^* with the $[\text{H}_2\text{O}\cdots\text{OH}_2]^+$ configuration (Fig. 2b).” (see Page 9, Lines 171-176)** Of course, as of today we cannot completely exclude the possible existence of a competing mechanism for N_2 fixation by $[\text{H}_3\text{O}^+\cdots\text{OH}]$ structure. To reliably test that we would need another, most probably multireference, quantum chemical method (not B2GP-PLYP) (see response 4 and Page 28, Lines 569-582).

3) In the Reaction mechanism section, it would be helpful to give the Cartesian coordinates of all optimized structures (or at least intermediates and transition state) in the Supporting Information.

⇒ **We thank the reviewer for the suggestion. The Cartesian coordinates (XYZs) of all optimized structures are now provided in the Supplementary information. (see Pages 23-26, Lines 270-365 in Supplementary information)**

4) The geometries and electronic energies were obtained with B2GP-PLYP density functional. Since energy span can gain marked improvement via a single point energy strategy, it would be helpful to utilize CCSD(T) single point energies in Fig. 2. In alternative to this, the authors can also test relative electronic energy differences between CCSD(T) method and B2GP-PLYP functional.

⇒ We thank the reviewer for the suggestion. CCSD(T) single point energies at complete basis set limit (CBS) were calculated and used as the major energy estimate in the revised manuscript. Some limitations of their accuracy due to multi-reference character of the system – especially our TS structure – are now discussed in the “Theoretical calculations” section (see Page 28, Lines 569-582; Pages 3-5, Lines 74-121 in Supplementary information). We also added a figure with FOD analysis for the TS structure (Supplementary Fig. S3). We have tested relative electronic energy differences, and the following text has been added into the revised manuscript: “Consistent with the results from numerous recent benchmark studies, a difference of ≈ 0.10 eV was observed between CCSD(T)/CBS and B2GP-PLYP/aug-cc-pVQZ data with the exception of $(\text{H}_2\text{O})_2^+$ conformations.” (see Page 28, Lines 579-582)

Fig. S3. Fractional occupation number weighted electron density (FOD) that is plotted as an isosurface at $\sigma=0.005$ e Bohr⁻³ level. TPSS/def2-TZVP at 5000 K data shows the “hot” (strongly correlated) electron density regions delocalized on the proposed TS system ($N^{\text{FOD}} > 0.65$).

5) The choice of virtual orbital-dependent functionals was determined by a comparison with available experimental data in literature. The authors tested DFT functionals for singlet and triplet N_2 . Do the authors test DFT functionals for other reactant and products (water dimer radical cation, HNO , and NH_2OH) if possible?

⇒ We thank the reviewer for the question. We clarified our methodology of DFT functional choice in the revised manuscript: “The geometries of reactants,

intermediates and products were optimized with B2GP-PLYP density functional with quadruple basis set. The same functional was used for vibrational analysis and the Gibbs free energy estimation at 298 K. This virtual orbital-dependent DFT method was chosen based on its performance on a set of small, related molecules with available experimental or high-level electronic structure data – namely N_2 , NO, HNO and $(H_2O)_2^+$ ”. (see Page 28, Lines 569-575). Detailed comparison was added to Supplementary information (see Pages 5-7, Lines 122-157 in Supplementary information). In general, B2GP-PLYP is more accurate by a factor of ~2-3 than classical DFs for geometries and frequencies. But, “unfortunately, inclusion of virtual orbitals into DFT paradigm does not help to reproduce the CCSD(T) relative energies of different $(H_2O)_2^+$ conformations: B2GP-PLYP/def2-QZVPP value of 0.64 eV for energy difference between hemibonded and H-bonded structures is hardly comparable with our value of 0.33 eV at CCSD(T)/CBS level or 0.31 eV from Schaefer.”. (see Pages 5-7, Lines 122-157 in Supplementary information).

6) “The INT1* structure is then converted into the excited-state intermediate HONH-HNOH*** by the direct double-proton transfer through TS* structure (Fig. 2).” Is the new mechanism of N_2 fixation confirmed by intrinsic reaction coordinate (IRC) analysis? If yes, which method is used?

⇒ **We thank the reviewer for the question.** During our preliminary analysis – before we came to realize the importance of multireference effects – we have used IRC analysis at M06-2X/6-311+G(d,p) level. Extreme problems with convergence were observed. Notwithstanding, after a long period of testing different G16 settings, we were able to converge the reaction mechanism. However, today we are fully aware that a completely different QC approach would be needed in order to fully validate our M06-2X/6-311+G(d,p) findings. Therefore, we chose not to mention the preliminary results of our IRC analysis in the manuscript in order to avoid confusion.

7) "Transition state energy was estimated from the combination of B2PLYP and M06-2X data." This sentence is a little confusing.

⇒ **We thank the reviewer for the question.** This confusing sentence was removed from the manuscript upon revision.

8) Give references for the basis set used.

⇒ **We thank the reviewer for this comment. We have significantly updated our references on theoretical studies: the corresponding references for the basis sets have been added in the revised supplementary information (see refs 2-19 in Supplementary information).**

Reviewer #4 (Remarks to the Author):

The manuscript "Efficient catalyst-free N₂ fixation by water under ambient conditions" by Zhang and coworkers describes a reaction that can produce HNO and NH₂OH from N₂ via reaction with the water dimer radical cation. The authors claim that this reaction is a good way to "fix" N₂ and therefore may have important utility. However, there are some concerns with some of the claims in this article that need to be reconciled. These are listed below:

⇒ **We thank the reviewer for his/her positive and insightful suggestions. We have done a detailed revision in the revised manuscript according to the suggestions.**

1. The generation of HNO and NH₂OH from N₂ disproportionation via this chemistry is intriguing. However, the authors fail to mention that HNO (a potent electrophile) reacts rapidly with NH₂OH (a potent nucleophile) to give N₂ and 2H₂O. This rapid and favorable reaction could negate the utility of this chemistry as sources for either HNO or NH₂OH.

⇒ **We thank the Reviewer for pointing out this problem. Indeed, HNO can react with NH₂OH under certain conditions to form N₂ and 2H₂O. However, in our**

experiments, the experimental device in Fig. 3a is specially designed in such a way that the HNO and NH_2OH^+ products could be spatially separated upon collection. The positively charged NH_2OH^+ product is driven by electric field into collection plate at the cathode, while the neutral HNO product is carried with the N_2 stream to the bottle at the upper outlet of the reactor. Therefore, the experimental design effectively prevents the back reaction between HNO with NH_2OH to form N_2 and $2\text{H}_2\text{O}$. In response, we have added the content into the Results and discussion section (Pages 10-11, Lines 201-209). It reads “Also, these observations enabled special experimental design whereby the HNO and NH_2OH^+ products could be spatially separated upon collection: the positively charged NH_2OH^+ product is driven by electric field into collection plate at the cathode, while the neutral HNO product is carried with the N_2 stream to the bottle at the upper outlet of the reactor (Fig. 3a). This experimental design effectively prevents the back reaction between HNO with NH_2OH to form N_2 and $2\text{H}_2\text{O}$. Note that according to our calculations the back reaction between the HNO and NH_2OH products while in the gas phase (e.g., *in situ* near the discharge area) is hindered by the rather significant energy barrier of ca. 1.4 eV.”.

2. Also, HNO is a very fleeting species due to self-reactivity to give N_2O and water. Therefore, in experiments where this chemistry was used as a source of HNO (e.g., the cell work), it needs to be realized that HNO is present for only a very short period (if at all) – thus, prolonged incubation is really meaningless.

⇒ We are very thankful to the reviewer for raising this question. The reviewer is correct that HNO can self-react through dimerization. However, the rate of HNO dimerization is strongly dependent upon concentration, pH and other parameters. Herein we quantitatively measured HNO solution at the concentrations of 0.2 μM and 250 μM over the period of 2 days. At 0.2 μM , there was essentially no loss of HNO observed (Fig. S11). At 250 μM some loss of HNO was observed (< 50%), which is probably caused by HNO self-reaction. Our observations are consistent with the recent reports indicating

certain stability of HNO in aqueous solutions under physiologically relevant conditions (J. Am. Chem. Soc., 2011, 133, 17912-17922, Anal. Chem. 2008, 80, 1247-1254). The following text has been added to the revised manuscript in order to address the reviewer's comment: Noteworthy, while HNO is known to dimerize at high concentrations ⁶³, the collected HNO solutions displayed considerable stability over the period of at least two days (Supplementary Fig. 11). The stability of HNO product is due to its relatively low concentration (sub mM), which prevents significant dimerization of HNO in the collected solution. Our observations are consistent with the recent reports indicating long-term stability of HNO in aqueous solutions under physiologically relevant conditions ^{64,65}. (Page 24, Lines 475-479)

Fig. S11. The stability of collected HNO aqueous solution at micromolar level. a
0.2 μM HNO. **b** 250 μM HNO.

3. Moreover, considering the propensity for HNO to react with and inhibit thiol proteins, it is difficult to understand how pre-incubation with HNO will protect cells from H₂O₂ toxicity since much of H₂O₂ toxicity is thought to be combatted using thiol proteins. Indeed, H₂O₂ toxicity is also thought to involve modification of thiol proteins.

⇒ We thank the reviewer for this question. The following text has been added upon revision in order to address the reviewer's comment: "It was reported that H₂O₂ signaling via thiol modification may lead to radical-mediated cellular damage (i.e., lipid peroxidation, protein carbonyl formation, etc.) due to the Fenton reaction ³³. In contrast, HNO demonstrates the ability to modify biological thiols, provide neuroprotection, and inhibit deleterious radical chain reaction, oxidative stress and

lipid peroxidation⁶⁷. Thus, HNO could interact with thiols thereby reducing the toxic effect of H₂O₂. In order to explore the possibility of using HNO to protect cells against H₂O₂ toxicity, we incubated HT22 cells with HNO and then treated them with H₂O₂.” (Page 25, Lines 496-503) Our results demonstrated that HNO can exert a protective effect against H₂O₂-induced oxidative stress and cell death in HT22 cells (Fig. 4f). This is consistent with the reference reported, i.e., HNO can serve as an antioxidant (Free Radical Biol. Med. 2007, 42, 482-491; Trends Pharmacol. Sci. 2008, 29, 601-608).

4. Also, it is interesting that HNO treatment of cells results in increased “viability”. The assay used actually measures the reducing capacity of the cell (not specifically viability). Although cell viability and reducing capability may correlate, this is not strictly viability. Regardless, some comment on what HNO may be doing is warranted here, especially since this is an unexpected finding.

⇒ We thank the reviewer for raising these comments. As per the reviewer’s comment, “viability” has been replaced with the more accurate “reducing capacity”. Also, we have incorporated the hypothesized mechanisms underlying our observations into the Results and discussion section (Pages 14-15, Lines 279-292). It reads “In model experiments, we demonstrated the potential utility of the collected HNO product in promoting the proliferation of HT22 cells and protecting them from H₂O₂-induced oxidative stress (Fig. 4e, 4f, Supplementary Fig. 10, 11). While the low dose of HNO (0.008-1 μM) remarkably increased cell reducing capacity, as measured by Cell Counting Kit-8 (CCK8), a high dose of HNO (1-32 μM) was found to inhibit cell reducing capacity (Fig. 4e, Supplementary Fig. 10). These observations suggest an important role of HNO in regulating cell growth³². It has been reported that HNO can interact with and modify several protein targets, such as thiol proteins, stimulating cell proliferation³³. In addition to that, through the reaction with the dehydrogenase and tetrazolium salts, HNO could also convert to NO, which further regulates cell growth. Indeed, NO has been established as a potent modulator of cell proliferation and the cell cycle, including stimulatory and

inhibitory effects^{34,35}. We propose that the observed two-phased effect of HNO on cell reducing capacity could be attributed to a complex signaling pathway involving NO. Further investigation is required to clarify the effects of HNO on cell reducing capacity.”.

Reviewer #5 (Remarks to the Author):

In their manuscript entitled "Efficient catalyst-free N₂ fixation by water under ambient conditions", Xiaoping Zhang et al. report the possibility to form nitroxyl and hydroxylamine by the reaction of N₂ with water dimer radical cations generated by plasma. The products are first observed by mass spectrometry and the process has then been transferred to a larger array-like reactor for wet chemical analysis. A possible reaction mechanism that sounds overall exotic but reasonable to me is derived with support from theoretical calculations. First of all, I can say that I read this manuscript with high interest and it is well written. Plasma-assisted N₂ fixation has been studied for a long time but I think that what is shown in the present submission is indeed a novel reaction pathway in this field (without plasma from N₂) which could be energetically more favourable than the state-of-the art methods as it is based on water vapor plasma. On the other side, the results shown here are that unexpected and unconventional that more control experiments are needed to fully support the conclusions. For the large related community working on electrocatalytic and photocatalytic N₂ fixation a few benchmark numbers are also missing that should be added. I recommend that the following experiments should be done and points should be clarified before possible publication in Nature Communications:

⇒ **We appreciate the reviewer’s positive and insightful suggestions. We conducted control experiments (the use of synthetic air experiment, ¹⁴N₂ and ¹⁵N₂ NMR experiments) as suggested. The results strongly supported the proposed reaction mechanism (see the following responses to single comments in detail) and supported our conclusion. In addition, we have provided some benchmark**

numbers of N₂ fixation (typically used in electrochemical and photocatalytic N₂ fixation) **such as conversion rate of N₂, NH₃ yield, comparison of energy cost, etc. to better demonstrate characteristics of N₂ fixation by our approach. We hope a clear and well-organized revised manuscript fully address all the concerns raised by the reviewer.**

1) In the plasma community the "conversion rate" of N₂ (in %) is an often used benchmark number. How is this conversion rate in this case?

⇒ We thank the reviewer for pointing out this question. The conversion rate of N₂ in this work is about 0.001%, which is higher than other N₂ plasma methods. The calculation was based on the formula reported by Gorbanev et al. (ACS Sustainable Chem. Eng., 2020, 8, 2996-3004). The manuscript has been revised in order to address this point as follows: “By integrating the calculated yields of N-containing products, the conversion rate of N₂ under the optimum conditions was estimated as ca. 0.001%, which is higher than in other plasma methods (for details see Supplementary information)³⁶. We believe that the higher conversion rate in our method is mainly due to the formation of abundant (H₂O)₂⁺ cations, which act as efficient activators of N₂ molecule.”. (Page 16, Lines 313-319)

Supplementary Information for details on the calculation of N₂ conversion rate:

Conversion rate = $\frac{C(\text{NH}_3 + \text{NH}_2\text{OH} + \text{NO}_2^- + \text{NO}_3^- + \text{HNO}) \text{ mol/L}}{10 \text{ min}} \times \frac{6 \text{ mL}}{1000 \text{ mL/L}} \times 24.5 \text{ L/mol} \times \frac{1}{\text{FR L/min}} \times 100\%$, where FR is the flow rate of the gas with 0.1 L/min. This is based on the assumption that all N₂ fixation products are converted to NH₂OH, HNO, NH₃, NO₂⁻ and NO₃⁻. In detail, C(NH₂OH) = 0.019 mmol/L, C(HNO) = 0.018 mmol/L, C(NH₃) = 0.02 mmol/L, C(NO₂⁻) = 0.004 mmol/L, C(NO₃⁻) = 0.007 mmol/L. (Page 2, Lines 23-29 in Supplementary information)

2) It is good that the reaction was performed in a scale up system where standard wet-chemical analysis becomes possible. I would expect authors to also perform these

experiments with ^{15}N and do NMR analysis of the formed products with a quantitative comparison of products formed from ^{15}N and ^{14}N .

⇒ We thank the reviewer for this suggestion. $^{14}\text{N}_2$ and $^{15}\text{N}_2$ NMR experiments were carried out (Fig. S7 below). A quantitative comparison was made, the results being close to the quantitative results of ion chromatography method. The following content has been added to the revised manuscript: “Further, the proton nuclear magnetic resonance (^1H NMR) spectrum of the condensate collected in the reaction between N_2 and water plasma displayed a peak at 10 ppm, which matched the peak generated by standard NH_2OH (Supplementary Fig. 8a). Consistently, the experiments with $^{15}\text{N}_2$ instead of $^{14}\text{N}_2$ displayed the characteristic $^{15}\text{NH}_2$ peak (Supplementary Fig. 8a).” (Page 13, Lines 248-251) and “Interestingly, ion chromatography (Supplementary Fig. 15) and ^1H NMR spectroscopy (Supplementary Fig. 7b) data also indicated the production of NH_4^+ (about 0.02 mM over 10 min), which is likely through the reduction of NH_2OH product. The simple reduction of NH_2OH to NH_3 is demonstrated by the results in Fig. 4a. The origin of NH_3 via the reduction of N_2 by water plasma was further confirmed by isotope-labeling experiments (Supplementary Fig. 7b).” (Page 16, Lines 307-312)

Fig. S8. NMR spectroscopy of reaction products produced using $^{14}\text{N}_2$ and the $^{15}\text{N}_2$ gas. a NH_2OH product. **b** NH_4^+ product. **c** Reaction device with enclosure. In order to increase the efficiency of product formation and reduce the cost of isotope $^{15}\text{N}_2$ gas, the $^{15}\text{N}_2$ gas was introduced into an enclosed container (Supplementary Fig. 8c). For comparison, the reference experiment with $^{14}\text{N}_2$ gas was done under the same conditions. The ^1H NMR spectrum of $^{14}\text{NH}_4^+$ product exhibited a typical triple peak with $J = 52$ Hz, which matched the peak generated by a standard NH_4Cl . When the mixture of $^{15}\text{N}_2/^{14}\text{N}_2$ or pure $^{15}\text{N}_2$ was used as the feeding gas, a double peak with $J = 72$ Hz was observed, which was assigned to the $^{15}\text{NH}_4^+$ product, similar to that reported by Hu et al ²¹.

3) N_2 with 99.99% purity is not very pure. It would be an expected reference experiment to simply use a 99.999% nitrogen source for comparison. Another very much recommended reference experiment would be the use of synthetic air.

⇒ We apologize for the omission. The purity of N₂ in this work is 99.999%. As suggested, the reference experiments with synthetic air were performed (Fig. S7 below) and we have incorporated the experiment results into the Results and discussion section (Pages 12-13, Lines 242-247). It reads “A series of reference experiments further confirmed the identify of NH₂OH product and its formation by N₂ reduction. No NH₂OH signal was detected either in the blank control setup or in the Ar atmosphere (N₂ was replaced by Ar in Fig. 3a), while clear NH₂OH signal was detected in both the synthetic air atmosphere and pure N₂ atmosphere (Supplementary Fig. 7). The efficiency of NH₂OH product formation in synthetic air was about 50% of that in N₂, which may be due to the lower content of N₂ in the synthetic air as well as due to the influence of O₂.”.

Fig. S7. Reference UV-Vis spectra recorded for the NH₂OH samples collected using the scaled-up setup shown in Fig. 3a under different conditions. The reference measurements show that the NH₂OH and HNO products were generated specifically due to the reaction between N₂ and (H₂O)₂⁺. The blank solution was 8-quinolinol solution (6 mL) exposed to humidified Ar under the same experimental conditions for 10 min.

4) In my opinion the chapter about the "Application of the reaction products" is not needed. The first part about ammonia formation might be but the rest should be removed which would authors give space for more experimental validation of their proposed mechanism.

⇒ We thank the reviewer for this suggestion. More experimental validation of the mechanism has been added in the revised manuscript. For the $^{14}\text{N}_2$ and $^{15}\text{N}_2$ NMR experiments, please see response 2 above. For the experiments of “the use of synthetic air”, please see response 3 above. For the detection for side reaction products, please see response 5 below. As other reviewers considered the section “Application of the reaction products” interesting and important, we chose to keep it in the main text albeit with some reductions: the formation of ammonia was retained, and the description of cell related content was mostly moved to Supplementary information and Methods section.

5) What would a FE of 20% mean in this case? Is there any side product of the reaction? I am also missing a comparative discussion of the overall energy input that would be needed to make, say, a mol of ammonia and the comparison to other plasma-based methods as well as H-B and electrochemical/photochemical N_2 fixation.

⇒ We thank the reviewer for the questions. The FE of 20% for HNO (nitrogen oxidation reaction) and FE of 64% for NH_2OH (nitrogen reduction reaction) indicate that there should be some side products in the reaction. Side products such as H_2O_2 , NO_2^- , NO_3^- and NH_3 were detected in the liquid phase. The concentration of H_2O_2 , NO_2^- , NO_3^- and NH_3 were calculated about 0.03 mM, 0.004 mM, 0.007 mM, and 0.02 mM, respectively. The corresponding experiment results are shown in Figures below. The discussion of the overall energy input in comparison with different methods has been added. The content of “Thus, in supplementary mass spectrometry (Supplementary Fig. 13a-d), ion chromatography (Supplementary Fig. 13e-f), and UV-vis absorption spectroscopy (Supplementary Fig. 14) we also observed NO_3^- , NO_2^- and H_2O_2 , products, which were likely produced by further conversion of the HNO product. The content of NO_3^- , NO_2^- and H_2O_2 produced over 10 min was estimated to be about 0.007 mM, 0.004 mM, and 0.03 mM, respectively. Interestingly, ion chromatography (Supplementary Fig. 15) and ^1H NMR spectroscopy

(Supplementary Fig. 7b) data also indicated the production of NH_4^+ (about 0.02 mM over 10 min), which is likely through the reduction of NH_2OH product. The simple reduction of NH_2OH to NH_3 is demonstrated by the results in Fig. 4a. The origin of NH_3 via the reduction of N_2 by water plasma was further confirmed by isotope-labeling experiments (Supplementary Fig. 7b).” **has been added in the revised manuscript. (Pages 15-16, Lines 302-312)**

⇒ **In method section,** “Quantification of NH_3 , NO_3^- and NO_2^- with ion chromatography. The amounts of NH_3 , NO_3^- and NO_2^- formed over the interaction between water plasma and N_2 were also estimated by ion chromatography (Thermo-Fisher DIONEX ICS-1100). The test procedure followed the standard operation procedure provided by the vendor. Briefly, 25 μL of the resultant was injected into the chamber for separation. Delivery speed was 4 mL min^{-1} . The peak eluted at the retention time of 9 min was assigned to NH_4^+ . The amount of NH_4^+ was estimated using a calibration curve by standard solutions (Supplementary Fig. 15). The peaks eluted at the retention time of 5 min and 8 min were assigned to NO_2^- and NO_3^- . The amounts of NO_2^- and NO_3^- were estimated using a series of standard solutions (Supplementary Fig. 13e-f).” **and** “Quantification of H_2O_2 by UV-Vis spectroscopy. The amount of H_2O_2 was measured by UV-Vis absorption spectroscopy (UV-1900i, Shimadzu, Japan). According to reference⁶⁶, potassium permanganate was reduced by hydrogen peroxide in sulfuric acid with the maximum absorption peak at 525 nm. The amount of H_2O_2 was derived from the amount of consumed MnO_4^- for calibration with a series of standard solutions (Supplementary Fig. 14).” **have been added in the revised manuscript. (see Pages 24-25, Lines 481-495)**

⇒ **The comparison of the overall energy input with different methods has been added. It reads** “In comparison with the HB process and other methods of N_2 fixation, including catalytic and plasma methods (Tables 1 and 2), our method offers considerably higher economy efficiency (1.54 \$ kwh^{-1} for NH_2OH and 7310 \$ kwh^{-1} for HNO), which is owing to the high value of NH_2OH and particularly

HNO products as compared to NH_3 . In terms of energy cost (g kWh^{-1}), our method is currently ca. 3 orders of magnitude less efficient than HB and ca. 1-2 orders of magnitude less efficient than plasma-based methods for N_2 fixation reported earlier (0.154 g kWh^{-1} for NH_2OH and 0.149 g kWh^{-1} for HNO). The higher energy cost in our method is mainly due to the avoidance of a catalyst, high pressures and high temperatures. The energy cost is expected to reduce as the plant continues to be upgraded. In comparison with several catalytic methods for N_2 fixation under mild conditions (near room temperature and atmospheric pressure) reported earlier, our method offers similar yield of N-containing products ($1.14 \mu\text{g}\cdot\text{cm}^{-2}\cdot\text{h}^{-1}$ for NH_2OH and $0.37 \mu\text{g}\cdot\text{cm}^{-2}\cdot\text{h}^{-1}$ for HNO).” (see Pages 16-17, Lines 320-331)

Fig. S13. Characterization of NO_2^- and NO_3^- products by mass spectrometry (a-d)

and ionic chromatography (e-h): **a** online detection of produced species in negative ion detection mode. **b** reference spectrum of standard compounds. **c** blank reference spectrum; **d** calibration curve for the quantitative evaluation of NO_2^- and NO_3^- products. **e** ion chromatogram of the collected reaction gas. **f** ionic chromatogram of standard compounds. **g** reference blank ion chromatogram. **h** calibration curve.

Fig. S14. Characterization of H_2O_2 product by UV-Vis spectroscopy.

Fig. S15. Characterization of NH_4^+ by ionic chromatography.

6) What would be the estimated ammonia production rate?

⇒ We thank the reviewer for this question. The ammonia production rate was estimated as $\sim 1.8 \mu\text{g cm}^{-2} \text{h}^{-1}$. As a response, the content of “As NH_2OH can be converted to NH_3 with nearly 100% efficiency, the estimated ammonia production rate under the optimum conditions was ca. $1.8 \mu\text{g cm}^{-2} \text{h}^{-1}$ (for details see Supplementary information).” has been added in the revised manuscript. (Page 16, Lines 317-319)

Supplementary Information for details on the calculation of NH_3 yields:

According to the detection of NH_4^+ by ion chromatography (Figure below), the

concentration of NH_4^+ is $\sim 0.36 \mu\text{g/mL}$. The NH_4^+ production rate was about $0.66 \mu\text{g cm}^{-2} \text{h}^{-1}$, which was calculated by the formula: $\text{Yield} = (C \times V) / (T \times S)$, where C is the concentration, V is the solution volume, T is the reaction time, S is the area of tip array. Because NH_2OH can convert to NH_3 with nearly 100% efficiency, the ammonia production rate was estimated as $\sim 1.8 \mu\text{g cm}^{-2} \text{h}^{-1}$. (Page 22, Lines 263-268 in Supplementary information)

Fig. S15. Characterization of NH_4^+ by ionic chromatography.

Other points:

- The title of the work will attract attention but should somewhere contain the word plasma. It is otherwise incomplete and misleading.

⇒ We thank the reviewer for the suggestion. The title has been changed to “Efficient catalyst-free N_2 fixation by water radical cations under ambient conditions” in the revised manuscript. We feel that this title well reflects the essence of our work and its relevance to plasma.

REVIEWER COMMENTS

Reviewer #1 (Remarks to the Author):

The authors have carefully addressed my comments in their revised manuscript. I especially appreciate the modifications and additions that have been made to make the study clearer and more complete. I do have some additional comments that I would like addressed before accepting for publication:

1. On Line 60 the authors state, "Recently, we have discovered that abundant radical cations of water clusters, especially in the dimer form, $(\text{H}_2\text{O})_2^+$, can be produced by depositing suitable amounts of energy into the pure water vapor at atmospheric pressure 17."

What does "energy" mean? Can you please be more specific? Are they referring to the energetic electrons formed in a low temperature plasma?

2. On Line 92, the authors have nicely clarified the different configurations for the synthesis: "Remarkably, when N_2 was flown into the discharge area (carrier argon gas was replaced by N_2 : Fig. 1a), abundant ions at m/z 32 and m/z 33 and m/z 64 were additionally observed (Fig. 1b). Note that the same signals were observed at similar intensity when N_2 was introduced to intersect with the water plasma ca. 1 cm away from the discharge area (Supplementary Fig. 1c, 1d)."

I really appreciate the control experiments. But I am surprised and unclear how N_2 (with water vapor) in the plasma gives the same results as N_2 introduced separately from the water vapor.

First of all, when there is no N_2 and only water vapor in the plasma for the control experiment (Suppl. Fig. 1c) is the N_2 still going through a plasma (a second plasma)? Is the plasma identical? How far is the end of the electrode/plasma from the mixing point with the water vapor?

Second, when N_2 is in the plasma, is the N_2 excited the same as when it is separate? Was any spectroscopy done to check? Were there any different excited species formed in the plasma? Was there any difference from the experiments where N_2 is introduced 1 cm downstream? Were the products measured by MS more abundant? Were there perhaps other products not measured by MS generated?

Third and finally, what is the difference between the experiment where N_2 is introduced 1 cm downstream of the plasma, and the ion trap experiment? The authors have explained the difference from when N_2 is mixed with the water vapor in the plasma, but not these two experiments where N_2 is added after. For me, it just confuses the results and mechanism even more. Are they trying to check that the ion trap in the MS doesn't affect the detection of products made in the plasma? It would be helpful to motivate the purpose of the ion trap experiment.

3. I thank the authors for now showing characterization and confirmation of other nitrogen products such as NO_3^- , NO_2^- , and NH_4^+ . Now what is missing is a comparison of these products in terms of mass

amount (production rate, conversion, etc.) and energy consumption to the main products of interest, NH₂OH and HNO. This is especially important because NH₂OH and HNO are intermediates and could be converted into NH₃ and NO₂⁻ or NO₃⁻. Thus, to judge their process, a comparison needs to be made. This will show how well the intermediates, NH₂OH and HNO, are quenched/stabilized, and how their overall mass selectivity and energy usage look, and if there needs to be future optimization to increase the selectivity and energy efficiency.

The above issue is also related to the response to Reviewer #4 and the stability of NH₂OH and HNO. The authors have also responded partly to my question to Reviewer #5 and the FE. But I would like the mass and energy values to be clearly shown in the paper, maybe as a figure or a table. One possibility is to add to Tables 1 and 2. I think showing only NH₂OH and HNO as products is misleading. The other major products should also be indicated. Otherwise, it should be in a separate figure/table.

Minor comment:

In Fig. S13, the authors say “ionic chromatography” when it should be “ion chromatography”. Please also check and correct throughout.

Reviewer #3 (Remarks to the Author):

The authors have addressed all the issues I suggested, I recommend it publication in Nature Communications.

Reviewer #4 (Remarks to the Author):

The authors have for the most part satisfactorily responded to my original concerns regarding the chemistry associated with the products of this process, NH₂OH and HNO. However, in their response to one point regarding the difficulty in reconciling HNO protection from H₂O₂ toxicity since both reagents modify thiols, I suggest the authors modify the passage in Line 498, page 25, by omitting the line “.....ability to modify biological thiols,...”. This is a minor point.

Reviewer #5 (Remarks to the Author):

I have now carefully checked the revised version of this work as well as the response of the authors to my initial comments. As the initially submitted version, also the revised manuscript is a nice piece of work. Authors have addressed all points raised carefully and made the corresponding control experiments. I am still of the opinion that the paragraph about the use of the products is not necessary

and that a title with the word "plasma" would be most appropriate but it might also be a subjective impression. After the revisions made, I can recommend publication of the work in Nature Communications in its present form.

Changes made in manuscript in response to referees' comments and questions (All page numbers, line numbers, etc. refer to the *revised version* of the manuscript)

REVIEWER COMMENTS

Reviewer #1 (Remarks to the Author):

Comments to the Author

The authors have carefully addressed my comments in their revised manuscript. I especially appreciate the modifications and additions that have been made to make the study clearer and more complete. I do have some additional comments that I would like addressed before accepting for publication:

⇒ **We thank the reviewer for his/her positive and insightful suggestions. We have done a detailed revision of the manuscript according to the suggestions.**

1. On Line 60 the authors state, “Recently, we have discovered that abundant radical cations of water clusters, especially in the dimer form, $(\text{H}_2\text{O})_2^+$, can be produced by depositing suitable amounts of energy into the pure water vapor at atmospheric pressure 17.”

What does “energy” mean? Can you please be more specific? Are they referring to the energetic electrons formed in a low temperature plasma?

⇒ **To address the reviewer’s comment, the sentence of “Recently, we have discovered that abundant radical cations of water clusters, especially in the dimer form, $(\text{H}_2\text{O})_2^+$, can be produced by depositing suitable amounts of energy into the pure water vapor at atmospheric pressure.” has been changed as “Recently, we have discovered that abundant radical cations of water clusters, especially in the dimer form, $(\text{H}_2\text{O})_2^+$, can be produced by electron stripping from neutral water molecules in a strong electric field of the energy-tunable corona discharge of the**

pure water vapor at atmospheric pressure.” **in the revised manuscript. (Pages 3-4, Lines 57-60)**

2. On Line 92, the authors have nicely clarified the different configurations for the synthesis: “Remarkably, when N₂ was flown into the discharge area (carrier argon gas was replaced by N₂: Fig. 1a), abundant ions at *m/z* 32 and *m/z* 33 and *m/z* 64 were additionally observed (Fig. 1b). Note that the same signals were observed at similar intensity when N₂ was introduced to intersect with the water plasma ca. 1 cm away from the discharge area (Supplementary Fig. 1c, 1d).”

I really appreciate the control experiments. But I am surprised and unclear how N₂ (with water vapor) in the plasma gives the same results as N₂ introduced separately from the water vapor.

First of all, when there is no N₂ and only water vapor in the plasma for the control experiment (Suppl. Fig. 1c) is the N₂ still going through a plasma (a second plasma)? Is the plasma identical? How far is the end of the electrode/plasma from the mixing point with the water vapor?

⇒ **We thank the reviewer for this question. We apologize for the somewhat confusing description in the original manuscript. In the two-channel experiment (Supplementary Fig. 1c), N₂ is introduced through a neutral Teflon tube (not as plasma) separately from the charged water vapor channel. Our control experiments show that the absolute intensity of product ions in the single-channel configuration (water vapor and N₂ both flown through the same charged channel) is about two-fold higher than in the two-channel configuration (see Fig. 1a,1b and Supplementary Fig. 1c,1d below). The word “similar” in the sentence pointed at by the reviewer was used sloppily and not accurately in the original manuscript. To address the reviewer’s comment, the content of “Remarkably, when N₂ was flown into the discharge area (carrier argon gas was replaced by N₂: Fig. 1a), abundant ions at *m/z* 32 and *m/z* 33 and *m/z* 64 were additionally observed (Fig. 1b). Note that the same signals were observed at similar intensity when N₂ was introduced to intersect with the water plasma ca. 1**

cm away from the discharge area (Supplementary Fig. 1c, 1d). These observations indicate that the signals m/z 32, m/z 33 and m/z 64 correspond to the species formed due to the interaction between water plasma and neutral N_2 .” **has been changed as** “Remarkably, when neutral N_2 was introduced to intersect with the water plasma ca. 1 cm away from the discharge area (Supplementary Fig. 1c), abundant ions at m/z 32 and m/z 33 and m/z 64 were observed (Supplementary Fig. 1d). This observation indicates that the signals m/z 32, m/z 33 and m/z 64 correspond to the species formed due to the interaction between water plasma and neutral N_2 . When N_2 was directly flown into the discharge area together with water vapor through the same channel (Fig. 1a), the same product ions at m/z 32 and m/z 33 and m/z 64 were observed with ca. two-fold higher intensity (Fig. 1b). The higher intensity of product ions in the single-channel configuration is probably due to the higher density of N_2 and water plasma right at the end of the electrode than ca. 1 cm away from the end of the electrode (as in the two-channel configuration), which results in higher collision rate between N_2 and water plasma species. Therefore, the single-channel configuration was applied in the subsequent scale up experiments to obtain higher yields of products.” **in the revised manuscript. (Page 6, Lines 92-105)**

[REDACTED]

Fig. 1 | Disproportionation reaction of N₂ with water dimer radical cation. **a** Experimental setup to study the interaction of N₂ with water vapor plasma at ambient conditions. Stainless-steel discharge needle was used as electrode. DC: direct current. **b** The corresponding mass spectrum of ionic products in Fig. 1a. Red-color marks correspond to the products specific to the reaction between water vapor plasma and N₂. **c** Ion trap reactor applied to study the reaction between N₂ and isolated (H₂O)₂⁺ (*m/z* 36) in vacuum. **d** The corresponding mass spectrum of ionic products in Fig. 1c. **e** Ionic fragments of the reaction intermediate at *m/z* 64 induced by collisional activation inside the ion trap. **f** Mass spectrum of the ionic species observed during the interaction between water vapor plasma and ¹⁵N₂ (¹⁵N₂ gas instead of ¹⁴N₂ in Fig. 1a). Red-color marks correspond to the products specific to the reaction between water vapor plasma and ¹⁵N₂.

[REDACTED]

Fig. S1. Reference experiments with regard to the reaction between N₂ and water radical cations shown in Fig. 1a. **a** Schematic illustration of the experiment in which N₂ in Fig. 1a was replaced by Ar. **b** Mass spectrum recorded in the experiments shown in Supplementary Fig. 1a was replaced by Ar. **c** Experimental two-channel setup to study the interaction of neutral N₂ with water radical cations by online mass spectrometry. **d** Mass spectrum recorded in Supplementary Fig. 1c. **e** Dissociation pathways of the HONH-HNOH⁺⁺ intermediate m/z 64 formed under the conditions of Fig. 1a. **f** Mass spectrum recorded when H₂O in the experiments shown in Fig. 1a was replaced by D₂O, other experimental conditions being the same. **g** Tandem mass spectra of the HONH-HNOH⁺⁺-type intermediates revealed in the disproportionation reaction of N₂ with (H₂O)₂⁺⁺ using different isotopic substitutes: m/z 68 (DOND-DNOD⁺⁺) and m/z 66 (HO¹⁵NH-H¹⁵NOH⁺⁺). **h** Mass spectrum of N₂⁺⁺ (corresponding signal at m/z 28) in the presence of neutral water vapor in the ion trap. No signals at m/z 64 (HONH-HNOH⁺⁺), m/z 32 (HNOH⁺) or m/z 33 (NH₂OH⁺⁺) could be observed under any experimental conditions tested, even by applying collision energy to the isolated N₂⁺⁺ ions (isolation width of 5 Da; the activation Q value of 0.4). **i** Isolation of ions at m/z 36 inside the ion trap with zero collisional activation energy.

Second, when N₂ is in the plasma, is the N₂ excited the same as when it is separate? Was any spectroscopy done to check? Were there any different excited species formed in the plasma? Was there any difference from the experiments where N₂ is introduced 1 cm downstream? Were the products measured by MS more abundant? Were there perhaps other products not measured by MS generated?

⇒ We thank the reviewer for these questions. Based on the results of reference

experiments, we believe that, whether in the plasma or separate, N_2 is excited the same way, i.e. through the collision with charged species at high kinetic energy. At this point we cannot propose any reasonable alternative to the proposed mechanism. Indeed, the amount of formed N_2^* in the two settings under discussion should be different due to the differences in ion/molecule densities, speeds and perhaps some other factors.

- ⇒ In present study no other spectroscopy techniques were used to characterize the plasma, mainly due to the lack of necessary instrumentation and engineering difficulties of interfacing these techniques with our setup. In our further research, we have planned a project to additionally characterize the excited species produced in water plasma by optical emission spectroscopy.
- ⇒ As far as the comparison with the experiments where N_2 is introduced 1 cm downstream, we did not observe any different products. The differences were mainly in the abundance of product ions measured by MS (more abundant in single-channel plasma), as discussed above.
- ⇒ In response, the content of “Based on the results of reference experiments in the Fig. 1c (N_2 is introduced in the plasma) and Fig. S1c (N_2 is introduced 1 cm downstream), we believe that, whether in the plasma or separate, N_2 is excited the same way, i.e. through the collision with charged species at high kinetic energy. At this point we cannot propose any reasonable alternative to the proposed mechanism. Indeed, the amount of N_2^* formed in the two settings under discussion should be different due to the differences in ion/molecule densities, speeds and perhaps some other factors. As far as the comparison with the experiments where N_2 is introduced 1 cm downstream, we did not observe any different products. The differences were mainly in the abundance of product ions measured by MS (more abundant in single-channel plasma).” has been added to the supplementary Material. (Page 8, Lines 175-183 in SI)

Third and finally, what is the difference between the experiment where N₂ is introduced 1 cm downstream of the plasma, and the ion trap experiment? The authors have explained the difference from when N₂ is mixed with the water vapor in the plasma, but not these two experiments where N₂ is added after. For me, it just confuses the results and mechanism even more. Are they trying to check that the ion trap in the MS doesn't affect the detection of products made in the plasma? It would be helpful to motivate the purpose of the ion trap experiment.

⇒ We thank the reviewer for this comment. Ion trap experiment is a key to support the mechanism shown in Fig. 2. In the ion trap experiment, out of the plethora of ionic species in water plasma, we were able to selectively isolate (H₂O)₂⁺ ions (*m/z* 36) and let them specifically interact with neutral N₂, no any other interfering species being present. The observation of the same species as in the plasma experiment (e.g., *m/z* 64, *m/z* 33) strongly supported the assumption that those species were formed through the reaction between (H₂O)₂⁺ and neutral N₂.

⇒ In response, the following text “To verify this assumption, (H₂O)₂⁺ ions (*m/z* 36) were selectively isolated in the ion trap by radio frequency (RF) field with peak-to-peak voltage of ~ 100 V in the presence of neutral N₂ gas, which was directly introduced into the ion trap (Fig. 1c).” **has been modified as follows:** “To verify this assumption, (H₂O)₂⁺ ions (*m/z* 36) formed in the water plasma were selectively isolated in the ion trap by radio frequency (RF) field with peak-to-peak voltage of ~ 100 V in the presence of neutral N₂ gas, which was directly introduced into the ion trap (Fig. 1c). This experimental design allowed us to specifically probe the intrinsic reactivity of N₂ toward (H₂O)₂⁺ in vacuum without any chemical interference.” **in the revised manuscript. (Page 8, Lines 107-112)**

3. I thank the authors for now showing characterization and confirmation of other nitrogen products such as NO₃⁻, NO₂⁻, and NH₄⁺. Now what is missing is a comparison of these products in terms of mass amount (production rate, conversion, etc.) and

energy consumption to the main products of interest, NH_2OH and HNO . This is especially important because NH_2OH and HNO are intermediates and could be converted into NH_3 and NO_2^- or NO_3^- . Thus, to judge their process, a comparison needs to be made. This will show how well the intermediates, NH_2OH and HNO , are quenched/stabilized, and how their overall mass selectivity and energy usage look, and if there needs to be future optimization to increase the selectivity and energy efficiency.

The above issue is also related to the response to Reviewer #4 and the stability of NH_2OH and HNO . The authors have also responded partly to my question to Reviewer #5 and the FE. But I would like the mass and energy values to be clearly shown in the paper, maybe as a figure or a table. One possibility is to add to Tables 1 and 2. I think showing only NH_2OH and HNO as products is misleading. The other major products should also be indicated. Otherwise, it should be in a separate figure/table.

⇒ **We agree with the opinion of the reviewer. We have added the mass and energy values of other products such as NH_3 , NO_2^- and NO_3^- in Tables 1 and 2 as suggested. To address the reviewer's comment, the content of "The product yield of NH_2OH was ca. two times higher than that of NH_3 , while the product yield of HNO was ca. three times lower than the integral product yield of NO_3^- and NO_2^- (Table 1). The lower yield of HNO compared to NO_3^- and NO_2^- probably reflects the high reactivity of HNO intermediate to give NO_3^- and NO_2^- . It is a challenge for further research to optimize the reaction parameters toward the higher yield and stability of the high-value HNO product." has been added in the revised manuscript. (Page 18, Lines 338-343)**

Minor comment:

In Fig. S13, the authors say "ionic chromatography" when it should be "ion chromatography". Please also check and correct throughout.

⇒ **Corrected throughout.**

Reviewer #3 (Remarks to the Author):

The authors have addressed all the issues I suggested, I recommend it publication in Nature Communications.

⇒ We thank the reviewer for the positive feedback.

Reviewer #4 (Remarks to the Author):

The authors have for the most part satisfactorily responded to my original concerns regarding the chemistry associated with the products of this process, NH_2OH and HNO . However, in their response to one point regarding the difficulty in reconciling HNO protection from H_2O_2 toxicity since both reagents modify thiols, I suggest the authors modify the passage in Line 498, page 25, by omitting the line ".....ability to modify biological thiols,...". This is a minor point.

⇒ We thank the reviewer's positive suggestions again. The content of "ability to modify biological thiols" has been deleted as suggested. (Page 27, Line 512)

Reviewer #5 (Remarks to the Author):

I have now carefully checked the revised version of this work as well as the response of the authors to my initial comments. As the initially submitted version, also the revised manuscript is a nice piece of work. Authors have addressed all points raised carefully and made the corresponding control experiments. I am still of the opinion that the paragraph about the use of the products is not necessary and that a title with the word "plasma" would be most appropriate but it might also be a subjective impression. After the revisions made, I can recommend publication of the work in Nature Communications in its present form.

⇒ We thank the reviewer for the positive feedback. We appreciate and see the rationale behind the reviewer's opinion that the paragraph about the use of the products may not be necessary and that a title with the word "plasma" would be most appropriate. However, we would prefer to retain the

application paragraph in the main text, given the novelty and particular value of NH_2OH and HNO products. Also, we prefer “water radical cations” over “plasma” for in our opinion it better reflects the scope and novel aspect of our study.

REVIEWERS' COMMENTS

Reviewer #1 (Remarks to the Author):

The authors have addressed my remaining comments. I thank them for their care and precise work. Congrats to them on the very nice contribution.

RE: NCOMMS-23-37005C: Efficient catalyst-free N₂ fixation by water radical cations under ambient conditions

REVIEWER COMMENTS

Reviewer #1 (Remarks to the Author):

Comments to the Author

The authors have addressed my remaining comments. I thank them for their care and precise work. Congrats to them on the very nice contribution.

⇒ **We thank the reviewer for the positive feedback.**